# Casposase structure and the mechanistic link between DNA transposition and spacer acquisition by CRISPR-Cas

**Alison B Hickman[1], Shweta Kailasan[1†], Pavol Genzor[2], Astrid D Haase[2], Fred Dyda[1]***

[1]Laboratory of Molecular Biology, National Institute of Diabetes and Digestive and Kidney Diseases, National Institutes of Health, Bethesda, United States; [2]Laboratory of Cell and Molecular Biology, National Institute of Diabetes and Digestive and Kidney Diseases, National Institutes of Health, Bethesda, United States

**Abstract** Key to CRISPR-Cas adaptive immunity is maintaining an ongoing record of invading nucleic acids, a process carried out by the Cas1-Cas2 complex that integrates short segments of foreign genetic material (spacers) into the CRISPR locus. It is hypothesized that Cas1 evolved from casposases, a novel class of transposases. We show here that the *Methanosarcina mazei* casposase can integrate varied forms of the casposon end in vitro, and recapitulates several properties of CRISPR-Cas integrases including site-specificity. The X-ray structure of the casposase bound to DNA representing the product of integration reveals a tetramer with target DNA bound snugly between two dimers in which single-stranded casposon end binding resembles that of spacer 3'-overhangs. The differences between transposase and CRISPR-Cas integrase are largely architectural, and it appears that evolutionary change involved changes in protein-protein interactions to favor Cas2 binding over tetramerization; this in turn led to preferred integration of single spacers over two transposon ends.

**\*For correspondence:**
Fred.Dyda@nih.gov

**Present address:** [†]Integrated BioTherapeutics, Inc, Rockville, United States

**Competing interests:** The authors declare that no competing interests exist.

## Introduction

CRISPR-Cas systems, encoded in the genomes of many bacterial and archaeal species, function as defense systems against invading foreign nucleic acids such as bacteriophages and plasmids (*Barrangou et al., 2007*; reviewed in *Mohanraju et al., 2016*). They provide acquired immunity in which the first step in thwarting an attack involves extracting short stretches of the foreign nucleic acid and archiving these DNA segments (or 'spacers') at the CRISPR locus where they alternate with host-specific repeats; this part of the process is known as adaptation (reviewed in *Marraffini, 2015*; *Sternberg et al., 2016*; *Amitai and Sorek, 2016*; *Jackson et al., 2017*). The CRISPR locus is then transcribed as a long CRISPR RNA (pre-crRNA) that is digested into shorter pieces (crRNAs). These 'guide' crRNAs are incorporated into a ribonucleoprotein interference complex that can recognize and cleave its complementary target sequence, thus destroying the invading nucleic acid.

Many different types of CRISPR-Cas systems have been discovered, with varying cohorts of associated genes and genomic architectures (*Makarova et al., 2018*). Common to all known autonomous CRISPR-Cas systems is the Cas1 protein (*Makarova et al., 2013*; *Makarova et al., 2018*; *Wright et al., 2019*) which acts in concert with its binding partner Cas2 to recognize, process, and integrate protospacers into the CRISPR-Cas locus (*Yosef et al., 2012*; *Nuñez et al., 2014*). Notable exceptions are the type V-C and V-D systems which lack a *cas2* gene (*Shmakov et al., 2015*; *Burstein et al., 2017*). Phylogenetic analysis of Cas1 proteins suggests that they may have originated from casposons, a novel class of putative DNA transposons in which a Cas1 homolog (the 'casposase') functions as the transposase (*Figure 1A*; *Krupovic et al., 2014*; *Koonin and Makarova,*

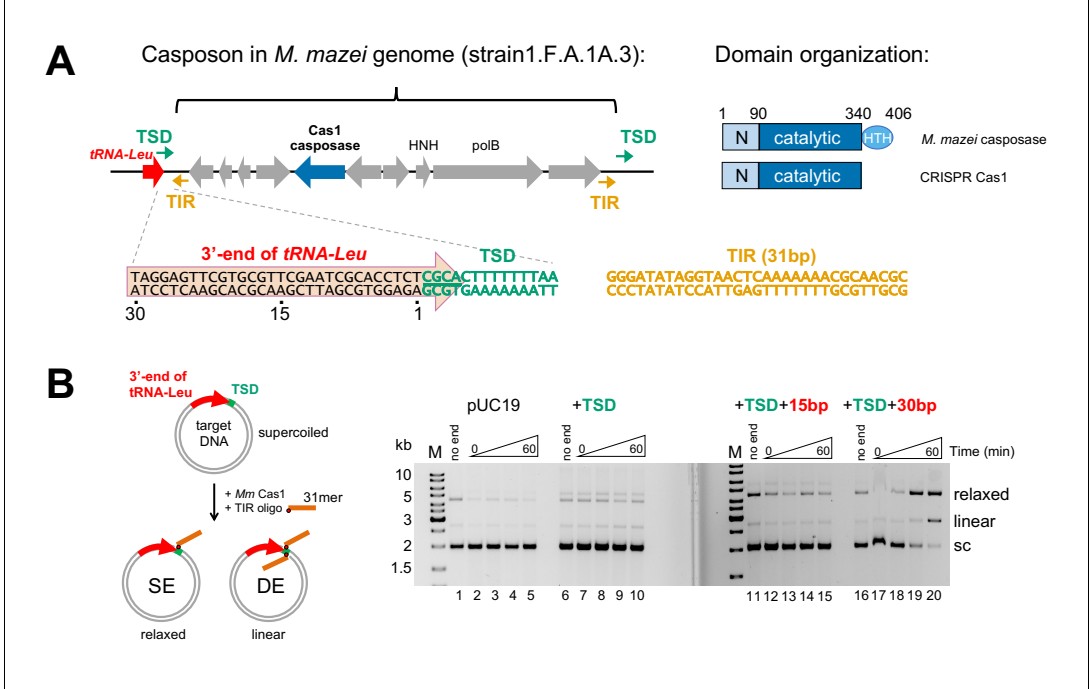

**Figure 1.** *Methanosarcina mazei* casposon organization and initial biochemical characterization. (**A**) (left) Schematic of a representative casposon from *Methanosarcina mazei* (the gene organization of casposon MetMaz1FA1A3-C2 is shown; *Krupovic et al., 2016*) adjacent to host tRNA-Leu gene (red arrow). The last 4 bp of the tRNA-Leu gene overlap with 4 bp of the 14 bp target site duplication (TSD; green arrows) that flanks the casposon. The casposon Terminal Inverted Repeats (TIR) are shown as orange arrows. *M. mazei* casposon genes are as annotated by *Krupovic et al. (2016)*, and are in gray with the exception of the putative Cas1-type casposase (in blue). Genes corresponding to a B-family type polymerase ('polB') and an HNH nuclease ('HNH') are marked. In the sequence below, a conserved 5'-CGCA motif at the 5'-end of the TSD is underlined. The sequence of the 31 bp TIR from strain S-6 and others, located at each casposon end, is shown in orange. (right) Relative to CRISPR Cas1, family two casposases have an additional C-terminal 60–70 residues predicted to form a helix-turn-helix (HTH) domain (*Krupovic et al., 2014*). Amino acid numbering is that of the *M. mazei* casposase. (**B**) In vitro integration assay of the *M. mazei* TIR into a target plasmid. The schematic at left shows the formation of relaxed and linear products corresponding to single-end (SE) and double-end (DE) integration of an oligonucleotide into a plasmid. The oligonucleotide is shown schematically in orange where the red dot indicates the 3'-OH nucleophile. In the agarose gels shown at the right, for the 'no end' controls, the plasmid was incubated for 60 min in the presence of protein and metal ion, but without the addition of the 31-mer TIR. For each reaction, the time points are: 0, 1, 5, and 60 min. Target: pUC19 (lanes 1–5); +14 bp TSD sequence (lanes 6–10); +TSD+15 bp more of the tRNA-Leu gene (lanes 11–15); +TSD+30 bp more of the tRNA-Leu gene (lanes 16–20).

*2017*). If so, the evolution of CRISPR-Cas systems would parallel that of the adaptive immune system in higher eukaryotes, which evolved from an ancient DNA transposon (*Kapitonov and Jurka, 2005*; *Carmona and Schatz, 2017*).

How Cas1-Cas2 complexes mediate spacer acquisition has been well-established through a series of genetic, biochemical, and structural studies (*Marraffini, 2015*; *Sternberg et al., 2016*; *Amitai and Sorek, 2016*; *Jackson et al., 2017*). The active integrase is an elongated $(Cas1)_2(Cas2)_2(-Cas1)_2$ heterohexamer in which two catalytic Cas1 monomers carry out metal-dependent transesterification to integrate protospacers into the target repeat sequence (*Yosef et al., 2012*; *Nuñez et al., 2014*; *Nuñez et al., 2015a*; *Nuñez et al., 2015b*; *Wang et al., 2015*; *Rollie et al., 2015*; *Xiao et al., 2017*; *Wright et al., 2017*). Efficient protospacer integration requires 3'-single stranded overhangs on each spacer end, and integration of the two 3'-ends into opposing strands of the target leads to the duplication of the CRISPR repeat.

Much less is known about casposases. Although the casposase from the archaeon *Aciduliprofundum boonei* is an active DNA integrase in vitro (*Hickman and Dyda, 2015*; *Béguin et al., 2016*; *Béguin et al., 2019*), in vivo transposition activity has yet to be demonstrated and how casposon ends competent for integration might be generated remains a mystery. Casposons contain varying collections of other genes (as shown schematically in *Figure 1A*), but only the casposase and a family B polymerase appear common to all (*Krupovic et al., 2017*). It is possible that casposases possess

an endonuclease activity that would generate the necessary free 3'-ends, but this has not yet been demonstrated (*Hickman and Dyda, 2015*). Alternatively, a cellular protein might be responsible for generating free casposon ends, by analogy to those proteins such as RecBCD or AddAB implicated in the generation of protospacers (*McGinn and Marraffini, 2019*). It has also been suggested that the casposon-encoded family B polymerase might be responsible for synthesizing linear copies of the casposon that could then be integrated (*Krupovic et al., 2017*).

Given the great abundance of DNA transposons and DNA transposase genes in nature - and in particular those with a catalytic core based on the RNase H-like fold (*Aziz et al., 2010*) - it is intriguing that a relatively sparsely distributed transposase was selected over the course of evolution to become the core of the CRISPR-Cas spacer acquisition system (*Krupovic et al., 2014*). Another puzzling aspect of the relationship is that, in most cases, DNA transposition involves strict specific recognition of transposon ends that are typically integrated into random locations in the genome whereas spacer acquisition involves the integration of many different fragments of invading DNA at a specific CRISPR locus just adjacent to a leader sequence.

To understand how Cas1 of CRISPR-Cas systems may have evolved from a transposase, we have characterized the biochemical properties of a casposase from *Methanosarcina mazei*. We have also determined its three-dimensional structure bound to branched DNA that represents the product of casposon end insertion into a specific sequence at the 3'-end of a host tRNA-Leu gene. We show that the casposase is tetrameric when bound to its target site and can site-specifically integrate a range of substrates including casposon ends with 3'-overhangs and even short single-stranded DNA ends as well as ssRNA. Our results suggest an evolutionary process in which a transposase that on its own recapitulates several canonical biochemical properties of CRISPR Cas1-Cas2 complexes acquired the ability to bind Cas2. This acquired interaction with a Cas2 dimer would have fundamentally changed the architecture of the integration complex, turning its properties away from transposon integration and towards spacer integration.

## Results

### The *Methanosarcina mazei* casposase is a site-specific integrase

The casposase from *A. boonei*, a species of archaea from the deep-sea hydrothermal vent euryarchaeota two lineage, is an active integrase that generates 14–15 bp target site duplications (TSDs) in vitro (*Hickman and Dyda, 2015*; *Béguin et al., 2016*; *Béguin et al., 2019*). It can integrate into random DNA but demonstrates a preference for site-specific integration into the very 3'-end of a host tRNA gene (*Béguin et al., 2016*; *Béguin et al., 2019*). Recently it was reported that a large family of closely related casposons are present in multiple strains of *Methanosarsina mazei* (*Krupovic et al., 2016*), an archaeal species that lives in semiaquatic and usually temperate habitats. Different strains of *M. mazei* have casposons in different genomic locations, strongly suggesting that these casposons are either currently active or have recently been active. One commonly observed genomic location for casposons in this species is within tRNA-Leu genes. Analysis of integration sites suggested that there is a short consensus sequence, 5'-CGCA, within the TSDs observed in vivo (*Krupovic et al., 2016*).

To determine if *M. mazei* casposase genes encode currently active integrases and whether they exhibit an insertion site preference, we expressed and purified a representative casposase (see Figure 6A), and tested its in vitro activity using an integration assay (*Figure 1B*). We initially used a 31mer dsDNA oligonucleotide ('dsLE31') which represents one of the casposon Left End (LE) sequences found in *M. mazei* strains; the particular sequence used (*Figure 1A*) is one of the shortest Terminal Inverted Repeats (TIRs) and is identical in multiple casposon copies in *M. mazei* (*Krupovic et al., 2016*). We used pUC19 and several variants of it as the target (*Figure 1B*). As *M. mazei* casposons have been frequently observed within tRNA-Leu genes (*Krupovic et al., 2016*), we elected to investigate one such possible target site. We first introduced 14 bp into pUC19 ('+TSD') corresponding to an observed genomic TSD comprised of the final 4 bp of the tRNA-Leu gene and 10 additional bp. The particular TSD sequence chosen (found in strain 1 .F.A.1B.3 and others, shown in green in *Figure 1A*) conforms to the in vivo consensus sequence in which a conserved 5'-CGCA motif is followed by an AT-rich sequence (*Krupovic et al., 2016*). We also generated pUC19 derivatives that contain either 15 or 30 more bp of the upstream tRNA-Leu gene (designated '+TSD+15

bp' and '+TSD+30 bp', respectively). Under the assay conditions used, the reaction time course shows that pUC19 itself is a poor target (lanes 1–5). Although there was little or no product formed when only the TSD (lanes 6–10) or TSD+15 bp (lanes 11–15) was introduced, relaxed products consistent with single-ended integration (SE) and linear products consistent with double-ended (DE) integration were readily observed when the TSD+30 bp of the tRNA-Leu gene were included (lanes 16–20). There was no further increase in activity when longer regions of the tRNA-Leu gene were included in the target plasmid (data not shown). These results suggested that integration by the *M. mazei* casposase is target-specific, and that the target site is contained within the sequence corresponding to TSD+30 bp. This is similar to the reported in vitro targeted integration of the *A. boonei* casposon into the 3' end of a host tRNA-Pro gene, although in that case, the data suggested that the target of this casposase is contained within a distinct TSD+15 bp region (*Béguin et al., 2019*).

To rule out the possibility that the casposon end oligonucleotide we used could be integrated using either of its terminal 3'-OH groups, we tested substrate derivatives with different phosphorylated 3' ends that would prevent their use as nucleophiles (*Figure 2A*). In this assay, we used a shorter casposon end oligonucleotide (dsLE17) which was also an effective substrate in the production of relaxed and linear products when incubated with the pUC19+TSD+30 bp target plasmid (lanes 1–4). It has previously been shown for the *A. boonei* casposase that ends considerably shorter than the entire TIR are competent for in vitro integration (*Hickman and Dyda, 2015*; *Béguin et al., 2019*). As shown in *Figure 2A*, relaxed and linear products were still formed when the remote 3'-OH group of dsLE17 was blocked by phosphorylation (lanes 5–7); conversely, when the nucleophilic 3'-OH group at the casposon end was blocked, there was no detectable product formation (lanes 8–10). In the context of the blocked casposon end, we also extended the top strand with 7-nt of ssDNA, and again no activity was detected (lanes 11–13). These results indicate that the casposase differentiates between the available 3'-OH groups of supplied oligonucleotides, and uses only the 3'-OH group at the casposon end as the nucleophile for integration.

As the precise nature of the casposon ends produced and used for integration is not known, we tested a variety of oligonucleotide substrates to determine if the *M. mazei* casposase demonstrates integration preferences similar to those of Cas1-Cas2 integrases (*Figure 2A,B*). Time courses were carried out in triplicate to provide quantitative information on the relative effectiveness of the substrates (*Figure 2—figure supplement 1*). Random oligonucleotides were included as controls of substrate specificity. One substrate we tested was TR17 (a 17mer casposon end which is recessed by five 5'-nt, '**T**op strand **R**ecessed', that is, there is a 5-nt 3'-overhang). This choice was motivated by the apparent preference of Cas1-Cas2 of some CRISPR-Cas systems to integrate spacers with short 3'-overhangs (*Nuñez et al., 2015a*; *Wang et al., 2015*; *Xiao et al., 2017*). We also included the single-stranded casposon left end bottom strand, ssLE17, as it has been shown by *Béguin et al. (2019)* that single-stranded casposon ends are effective substrates for the *A. boonei* casposase.

As shown in *Figure 2B* and *Figure 2—figure supplement 1*, with the exception of the random single-stranded 17mer ('ssran17') control, all oligonucleotide substrates tested produced relaxed and linear products consistent with single-ended and double-ended integrations when the specific target plasmid pUC19+TSD+30 bp was used, but not with pUC19. The level of production of the linear plasmid product (i.e., consistent with double-ended integration) was the highest for TR17; dsLE17 and ssLE17 produced similar intermediate levels, and ranTR17 and dsran17 were the least effective substrates.

We were intrigued by the capacity of the casposase to integrate ssDNA, and we repeated the assay with casposon-specific ssDNA oligonucleotides of varying length (*Figure 2C*). Remarkably, casposon ends as short as six nt could be specifically integrated, a result that has also recently been demonstrated for the *A. boonei* casposase (*Béguin et al., 2019*). Taken together, the results indicate that the *M. mazei* casposase has a wide tolerance for the type and sequence of oligonucleotides it can integrate, but with an apparent preference for a partially single-stranded casposon end. The putative evolution from transposition to spacer integration must therefore have involved acquiring the ability to discriminate among possible substrates and to integrate only short, recessed spacers of relatively specific length.

In order to quantify precise target site-specificity, we performed next-generation sequencing (NGS) of the linear plasmid products generated when pUC19+TSD+30 bp was used as the integration target (*Figure 3A–C*). Assays were carried out using either the 31mer casposon TIR (dsLE31) or a 29mer casposon end recessed by 5-nt on the top strand (TR29). As NGS has not been reported for

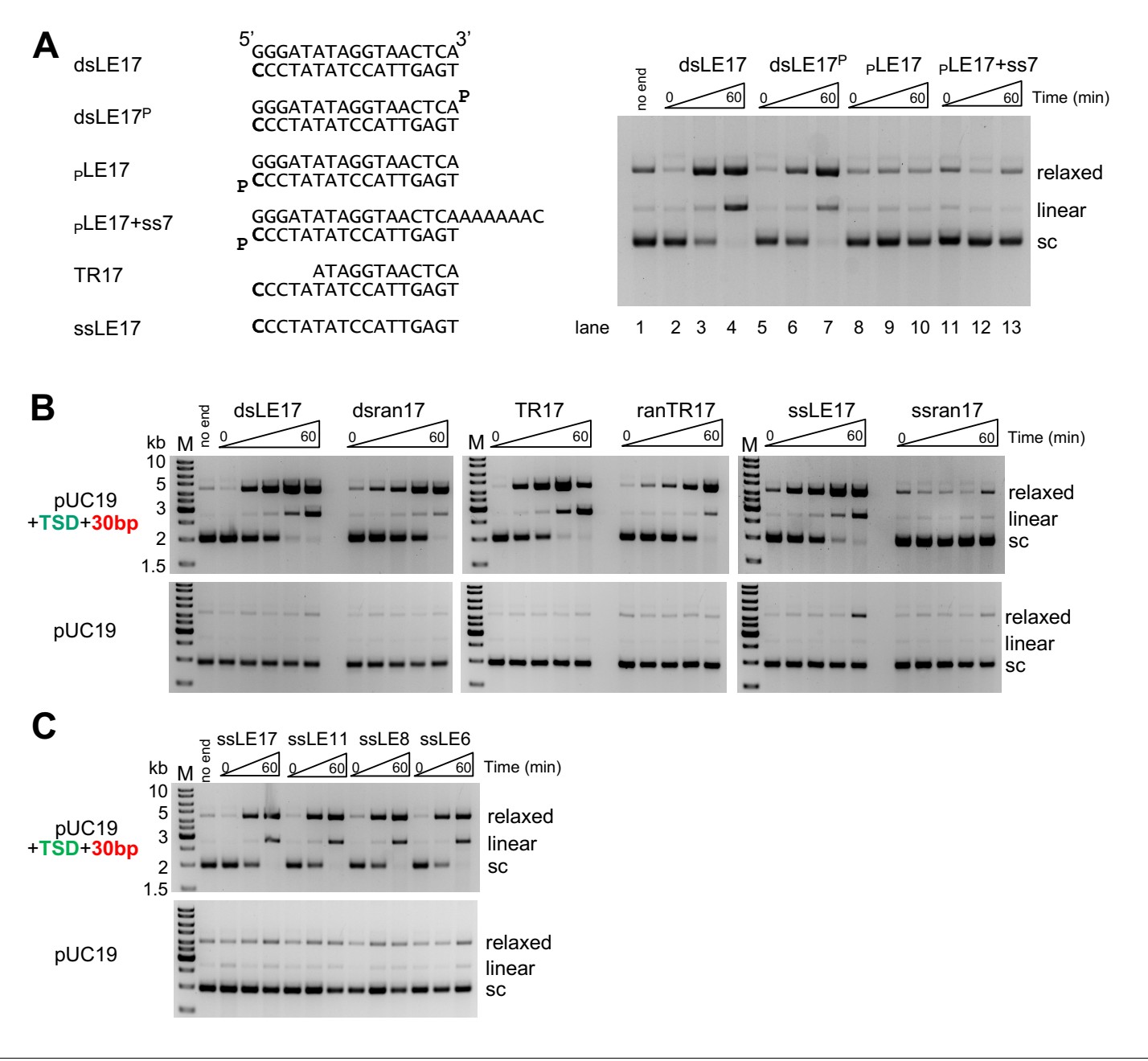

**Figure 2.** In vitro integration activity properties of the *M. mazei* casposase. (**A**) Effect of blocking free 3'-OH ends on product formation. (left) Sequences of TIR oligonucleotides used to assay integration into pUC19+TSD+30 bp. The 3'-end C at the casposon end is in bold. (right) For each reaction shown, time points are: 0, 1, and 60 min. (**B**) Integration into the specific target plasmid (pUC19+TSD+30 bp, top) compared to integration into pUC19 (bottom). Integration was assayed using a variety of substrate forms as shown (all sequences can be found in Table S1). For each reaction, time points are: 0, 0.5, 1, 5, and 60 min. Reactions were performed in triplicate, and product quantitation is in *Figure 2—figure supplement 1*. (**C**) Integration of single-stranded casposon ends. For each reaction, time points are: 0, 1, and 60 min.

The online version of this article includes the following figure supplement(s) for figure 2:

**Figure supplement 1.** Quantitation of time course of product formation.

casposases to date, we varied the sample preparation method in several ways. For four samples, linear reaction products resulting from the independent incubation with dsLE31 with pUC19+TSD+30 bp were isolated from agarose gels and subjected to either 17 cycles (duplicate samples S1A1, S2A1) or 30 cycles (duplicate samples S3A2, S4A2) of PCR prior to the preparation of sequencing

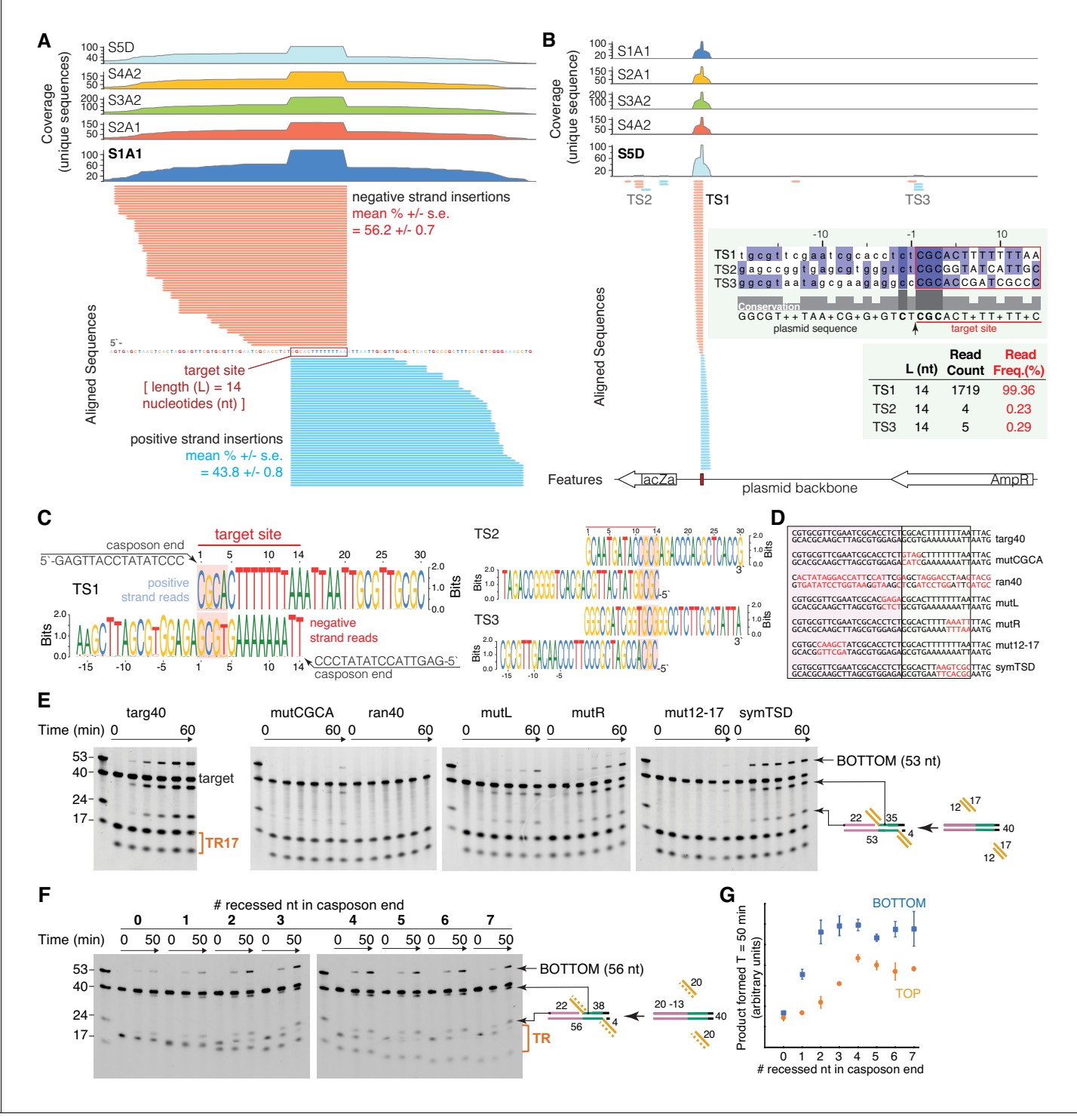

**Figure 3.** Characterization of target-site specificity. (A) Double-end integration products of five independent reactions (sample S1-S5) were purified and analyzed by next generation sequencing (NGS). Sample 1–4 were PCR-amplified before sample preparation for NGS. Sample five was directly subjected to NGS sample prep. The five independent reactions show the same dominant target site (TS1). The substrate sequence 3′ of the novel insertion identifies the 14 bp target-site. Reads were collapsed by unique sequences, clipped 3′ of the spacer sequence, and aligned to the target plasmid (pUC19+TSD+30 bp). Mean read frequencies (+ /- standard error) for insertions on the positive (blue) and negative (red) strand (A) and a representative sequence logo (C) are depicted. The negative strand logo has been flipped to visualize the sequence complementarity. The CGCA motifs are highlighted. (B) Sample S5D showed additional minor target sites (TS2 and TS3) with similar target motifs (C). (D) Target variants for assay of TR17 integration into a 40-mer target. Nucleotides in red indicate where targ40 sequences were mutated. (E) As shown schematically at the right,

*Figure 3 continued on next page*

*Figure 3 continued*

integration into the target top strand generates a 22-mer and 35-mer product pair; integration into the bottom strand generates a 53-mer (and a 4-mer too short to be detected). For each reaction, time points are: 0, 0.25, 1, 2, 5, and 60 min. The first lane in each gel contains standard size markers for reference. (F) Integration of a 20mer casposon end ('TR' on the right) into targ40 as a function of the number of recessed nucleotides on the top strand. The experiment is shown schematically at the right where the recessed strand is shown as the dashed orange line. Reactions were performed in triplicate, and quantitation of product formation (as measured as band intensity) for both the bottom strand (blue) and top strands (orange) is shown in (G). Plotted points represent the mean, and the error bars correspond to the standard deviation. ssRNA can also be integrated (*Figure 3—figure supplement 1*).

The online version of this article includes the following figure supplement(s) for figure 3:

**Figure supplement 1.** Integration of ssRNA into targ40.

libraries. We also carried out a large-scale reaction using TR29 so that enough linear product could be obtained directly to avoid the need for PCR amplification prior to library preparation (sample S5D). This allowed us to assess the relative integration frequency for the two possible orientations of the casposon end, as our PCR strategy would not detect products resulting from the use of the remote 3'-OH as the nucleophile.

The results reveal that integration occurred overwhelmingly and with precision into the specific target site of pUC19+TSD+30 bp (designated target site one or TS1), with sequences from all five samples revealing overlapping positive and negative strand insertions consistent with the 14 bp off-set corresponding to the size of casposon TSDs (*Figure 3A–C*). For the four samples generated using PCR amplification, no other integration sites were detected (*Figure 3B*). For sample S5D, 99.4% of the detected integration events were at TS1, and only two other sites in the pUC19 back-bone were detected with reads corresponding to integration into both strands of the target (TS2, TS3; *Figure 3B,C*). Alignment of the three sites (*Figure 3B*, inset) indicated that TS2 and TS3 contain a conserved 5'-CGC motif at one site of integration. Notably, all three observed target sites have a C nucleotide at the −2 position flanking the integration site but no other sequence conservation. We detected no sequences that would have been generated if the opposite 3'-end of the casposon end oligonucleotides were used as the nucleophile. Collectively, these results suggest that integration by the *M. mazei* casposase is remarkably sequence-specific, and that the most important target sequence features directing integration are clustered around the 5'-CGCA motif detected in vivo (*Krupovic et al., 2016*).

To explore aspects of target recognition by casposases in more detail, we employed an integration assay using short target oligonucleotides (*Figure 3D*) containing the region we established as sufficient for site-specific targeting. Integration into a 40mer target ('targ40') oligonucleotide by TR17, the most active substrate in the experiments in *Figure 2B*, was readily detectable (*Figure 3E*) and yielded products during the time course corresponding to the expected size for site-specific integration. The conserved 5'-CGCA target motif was critical as its mutation resulted in almost complete loss of both top and bottom strand integration ('mutCGCA'). We also established that there was no detectable integration into a random 40mer ('ran40'). Mutation of four bp just 5' of the CGCA motif ('mutL') severely reduced integration as did mutating the last five bp of the TSD ('mutR'), although to a lesser extent. As the plasmid assay had revealed that integration is dependent upon some portion of the upstream 15–30 bp tRNA sequence, we mutated the region corresponding to bp 12–17, and this abolished activity ('mut12-17'). We were also curious how symmetrizing the TSD (i.e., placing the 5'-CGCA motif on both strands at the target site) might impact activity and we found it was as effective a target as the original TSD ('symTSD').

We took advantage of this assay to more finely dissect the stimulatory effect of recessing the top strand of the casposon end (*Figure 3F*). Starting with a blunt-ended dsLE20 substrate, we progressively deleted single nucleotides from the top strand to thereby generate a series of substrates with progressively longer 3'-single-stranded overhangs. The assay was performed in triplicate and the product bands for integration into the top and bottom strands separately quantitated (*Figure 3G*). Under these conditions, there was little detectable product formation with the blunt-ended substrate and only slight stimulation with a 1-nt 3'-single-stranded overhang. Maximal activity for bottom strand integration was obtained once the 3'-overhang was 2-nt and was thereafter essentially unchanged as the length of the overhang was increased to 7-nt. Integration into the top strand as a function of 3'-overhang length lagged behind that of bottom strand integration, and reached its

maximal level once the 3'-overhang was 4-nt. Although it is not clear if the difference between the top and bottom strand is significant or a consequence of the asymmetry of the target, these results suggest that only a few unpaired nucleotides are required at the casposon end for optimal integration activity. It is possible that the reduced activity we observe with blunt-ended substrates reflects a need for transient fraying at the end for binding and integration.

Lastly, encouraged by the tolerance of the *M. mazei* casposase towards integration substrates and by the report of direct integration of ssRNA by a type III reverse transcriptase-Cas1 fusion protein (*Silas et al., 2016*), we also tested the RNA version of ssLE17. As shown in *Figure 3—figure supplement 1*, ssRNA also can be integrated into the target oligonucleotide targ40 although far less efficiently.

## *M. mazei* casposase is tetrameric when bound to target and its casposon ends

We next investigated the in vitro DNA binding properties of casposases using interferometric scattering mass spectrometry (iSCAMS, *Young et al., 2018*) and analytical size exclusion chromatography (SEC). At the low protein concentrations used for iSCAMS (25–50 nM), the *M. mazei* casposase had a measured MW of 95 kDa (*Figure 4*), consistent with a dimer and in keeping with previously characterized CRISPR Cas1 proteins and casposases (*Wiedenheft et al., 2009*; *Hickman and Dyda, 2015*). In the presence of a 17-nt ssDNA casposon end (ssLE17), the MW shifted to 109 kDa, suggesting binding while the protein remained dimeric. Due the uncertainty of this measurement, we could not distinguish between one or two casposon ends bound per dimer (104 kDa vs. 109 kDa). In the presence of a branched oligonucleotide substrate comprised of two single-stranded 8-nt

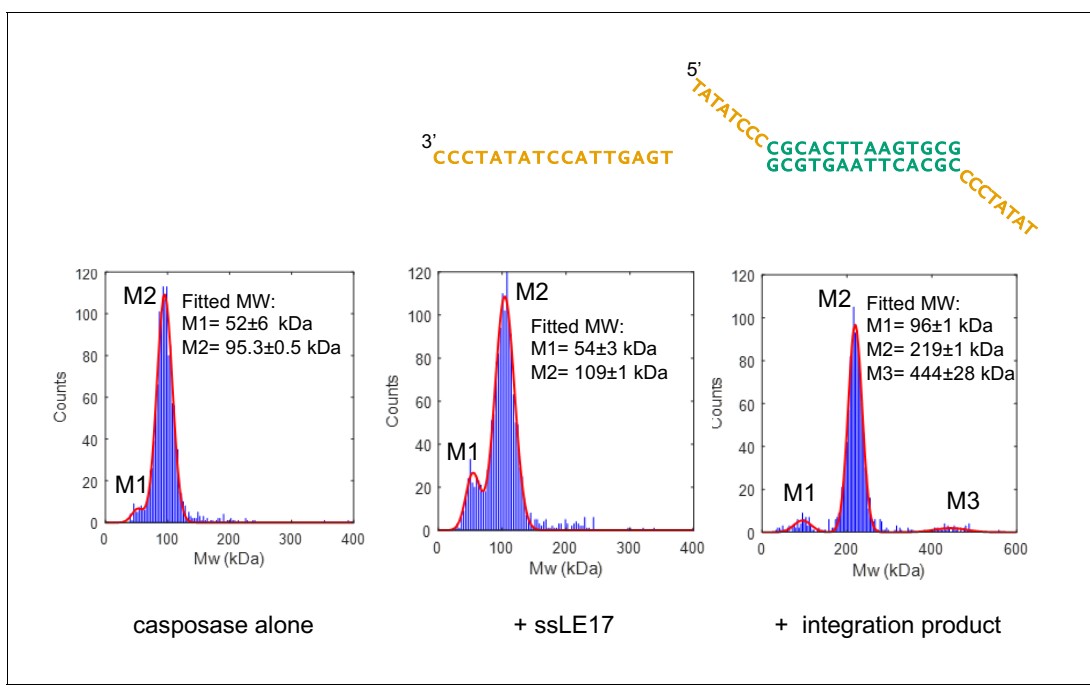

**Figure 4.** Characterization of casposase-DNA complexes. Characterization by interferometric scattering mass spectrometry (iSCAMS) analysis. Casposase alone (left; $MW_{calc}$ monomer = 49.4 kDa, dimer = 98.8 kDa), casposase bound to ssLE17 (center; $MW_{calc}$ dimer+ssLE17 = 104 kDa, dimer +2xssLE17 = 109.1 kDa), and casposase bound to integration complex used for crystallographic studies (right, $MW_{calc}$DNA = 13.3 kDa; tetramer +DNA = 210.9 kDa). Each experiment was performed once for each indicated binding substrate, and all of the data is included in the mass histograms. The reported error corresponds to the fitting error to a Gaussian distribution model. Characterization by size-exclusion chromatography (SEC) is shown in *Figure 4—figure supplements 1* and *2*.

The online version of this article includes the following figure supplement(s) for figure 4:

**Figure supplement 1.** Characterization of casposase-DNA complexes by size-exclusion chromatography (SEC) analysis.

**Figure supplement 2.** Characterization of casposase-DNA complexes by size exclusion chromatography (SEC) on a Superdex 200 Increase 3.2/300 column as a function of protein concentration.

casposon ends linked to a symmetrized target sequence mimicking the integration product (minus any overhangs that might be present in vivo), the resulting protein-DNA complex had a measured MW of 219 kDa, consistent with a tetramer bound to one integration product molecule.

We used SEC to resolve the uncertainty in the protein-DNA binding stoichiometry with casposon ends. When the casposase was titrated with either TR17 or ssLE17, the casposase bound with a stoichiometry of one end per dimer (*Figure 4—figure supplement 1*). During the course of these experiments, it became clear that the elution position of the casposase varied as a function of concentration (*Figure 4—figure supplement 2*) whereas its complexes with either TR17 or with the integration product studied by iSCAMS eluted at positions expected for a tetramer over a range of protein concentrations. Thus, it appears that the casposase exhibits a concentration-dependent equilibrium between dimeric and tetrameric forms, and that this equilibrium is affected upon binding to DNA substrates.

In an effort to further characterize the casposase and its interactions with its substrates, we attempted to crystallize the casposase bound to DNA oligonucleotides representing different stages of the integration reaction. Many casposase-DNA complexes yielded poorly-diffracting crystals including the casposase bound to recessed dsDNA oligonucleotides and ssDNA oligonucleotides of varying length representing the casposon ends. This was also the case with casposase bound to mimics of the post-integration step using both symmetric and asymmetric integration target sites and casposon ends of varying length and dsDNA-ssDNA composition. We were eventually able to obtain diffracting crystals of the casposase bound to a symmetric 14mer target site into which two single-stranded 8-nt casposon ends were integrated (the integration product shown in *Figure 4*), and the structure of the complex was solved to 3.1 Å using experimental phases obtained from a selenomethionine derivative (*Table 1*; *Figure 5*; *Figure 5—figure supplement 1*).

The pseudo two-fold symmetric complex is a clamp-like tetramer in which two casposase dimers hold the double-stranded target site between them, with the 5' termini of the single-stranded casposon ends emerging through a narrow slot in the bottom of the clamp (*Figure 5A,B*). Each casposase monomer exhibits the fold expected of a Cas1 homolog (*Wiedenheft et al., 2009*; *Figure 5—figure supplements 2* and *3*), with a canonical eight-stranded N-terminal domain and a globular catalytic domain that brings conserved active site residues (E162, H232, E247) into close proximity (*Figure 5C*). When compared to its most closely related structurally characterized Cas1 homolog, that from *Archaeoglobus fulgidus* (*Kim et al., 2013*) with which it shares 32% sequence identity, the N-terminal domains have an RMSD of 1.68 Å over 72 shared Cα positions and the catalytic domains are even more similar, with an RMSD of 1.28 Å over 208 shared Cα positions.

The casposase dimers that are the opposite sides of the clamp are each formed by the juxtaposition of two N-terminal β-strand domains into two extended eight-stranded β-sheets (*Figure 5—figure supplement 2*, top), as has been observed for CRISPR Cas1 dimers. In the integration product complex, each dimer is functionally and structurally asymmetric with a 'catalytic' monomer whose active site contains the scissile phosphate group that links transposon end DNA and the target site, and a 'noncatalytic' monomer in which the active site is located on the periphery of the assembly facing away from the DNA. This functional asymmetry at the dimer level recapitulates the observed asymmetry of Cas1 dimers in structurally characterized CRISPR Cas1-Cas2 integration complexes bound to DNA (*Nuñez et al., 2015b*; *Wang et al., 2015*; *Xiao et al., 2017*; *Wright et al., 2017*).

In the complex, there is interpretable density for all four monomers until amino acid 341, but thereafter the C-terminus is not visible in two of the monomers and, in the other two, there is interpretable density only until residues 354 and 359. We confirmed that the C-terminal residues are still present in the crystallized complex and the protein had not been proteolytically degraded (*Figure 5—figure supplement 4*). Close examination of solvent channels and difference electron density maps revealed density consistent with isolated α-helices that are likely regions of the C-terminal HTH domain but we are unable to assign them to a specific register. Thus it appears that when the casposase is bound to this integration product mimic, the C-terminal residues remain disordered within the crystal and do not stably engage with the rest of the protein or DNA. In an attempt to provide insight into the role of the C-terminus, we expressed and purified a C-terminally truncated version of the casposase consisting of residues 1–341 (ΔHTH, *Figure 6A*), and assayed its activity against the range of substrates previously evaluated for the full-length protein. As shown in *Figure 6B*, there was no evidence of integration activity with any of our integration substrates, using either the specific pUC19+TSD+30 bp target or on pUC19 alone. Thus, the casposase N-terminal

**Table 1.** X-ray structure determination detail.

| Data set | E1 | E2 | E3 | Refinement |
|---|---|---|---|---|
| Space group | $P4_32_12$ | $P4_32_12$ | $P4_32_12$ | $P4_32_12$ |
| Wavelength (Å) | 0.97930 | 0.97916 | 0.96802 | 0.97918 |
| Unit cell parameters | | | | |
| $a = b$, $c$ (Å) | 106.91, 424.61 | 106.98, 424.24 | 106.83, 423.30 | 106.99, 423.36 |
| $\alpha = \beta = \gamma$ (°) | 90, 90, 90 | 90, 90, 90 | 90, 90, 90 | 90, 90, 90 |
| Resolution range (Å) | 30.0–3.2 | 30.0–3.2 | 30.0–3.2 | 30.0–3.1 |
| No. of unique reflections | 77309 | 77464 | 77076 | 45362[a] |
| Redundancy | 7.08 | 7.13 | 7.19 | 14.52 |
| I/σI (3.29–3.2 Å) | 12.16 (1.28) | 12.45 (1.54) | 12.26 (1.93) | 13.3 (1.95) |
| $B_{Wilson}$ (Å$^2$) | 100.7 | 93.88 | 90.13 | 104.36 |
| Completeness (%) | 99.8 | 99.8 | 99.8 | 99.8 |
| $R_{merge}$ (%)[b] | 18.2 | 18.0 | 14.7 | 11.4 |
| CC1/2 (last resolution shell) | 49.2 | 60.8 | 77.0 | 81.9 |
| Phasing power (iso/ano) | 0.0/1.045 | 0.214/0.995 | 0.877/0.812 | |
| Figure of merit at 3.2 Å | 0.38 | | | |
| Structure Refinement | | | | |
| Resolution (Å) | | | | 3.1 |
| $R_{work}$/$R_{free}$ [c] | | | | 0.198/0.256 |
| RMSD | | | | |
| Bond lengths (Å) | | | | 0.01 |
| Bond angles (°) | | | | 1.17 |
| Average B factor (Å$^2$) | | | | 142.33 |
| No. atoms | | | | |
| Protein | | | | 11442 |
| DNA | | | | 846 |
| Ligands (Ca$^{2+}$) | | | | 6 |
| Waters | | | | 36 |
| No. of reflections | | | | 45362 |
| Ramachandran plot (%) | | | | |
| Favored | | | | 86.6 |
| Allowed | | | | 13.1 |
| Outliers | | | | 0.2 |

Numbers in parentheses represent the value for the highest-resolution shell. RMSD, root-mean-square deviation.

[a] Friedel's law true.

[b] $R_{merge} = \Sigma|I_i - <I>|/\Sigma I_i$, where $I_i$ is the intensity of measured reflection and $<I>$ is the mean intensity of all symmetry-related reflections.

[c] $R_{free} = \Sigma_T||F_{calc}| - |F_{obs}||/\Sigma F_{obs}$, where T is a test dataset of about 5% of the total unique reflections randomly chosen and set aside prior to refinement.

and catalytic domains alone are not sufficient for integration, and the HTH domain must play a crucial role.

In the tetrameric complex with the integration product DNA, we observe two dimer-to-dimer interactions (*Figure 6C*) formed by the convergence of helix α1 (residues 59–67) and a loop (residues 39–44 between β4 and β5) from each noncatalytic monomer with two loops (residues 182–199 and 275–278) from the opposing catalytic monomer. Each set of interactions buries a surface area of ~1200 Å in the tetramerization interface and together form the base of the clamp in which the dsDNA target is enclosed.

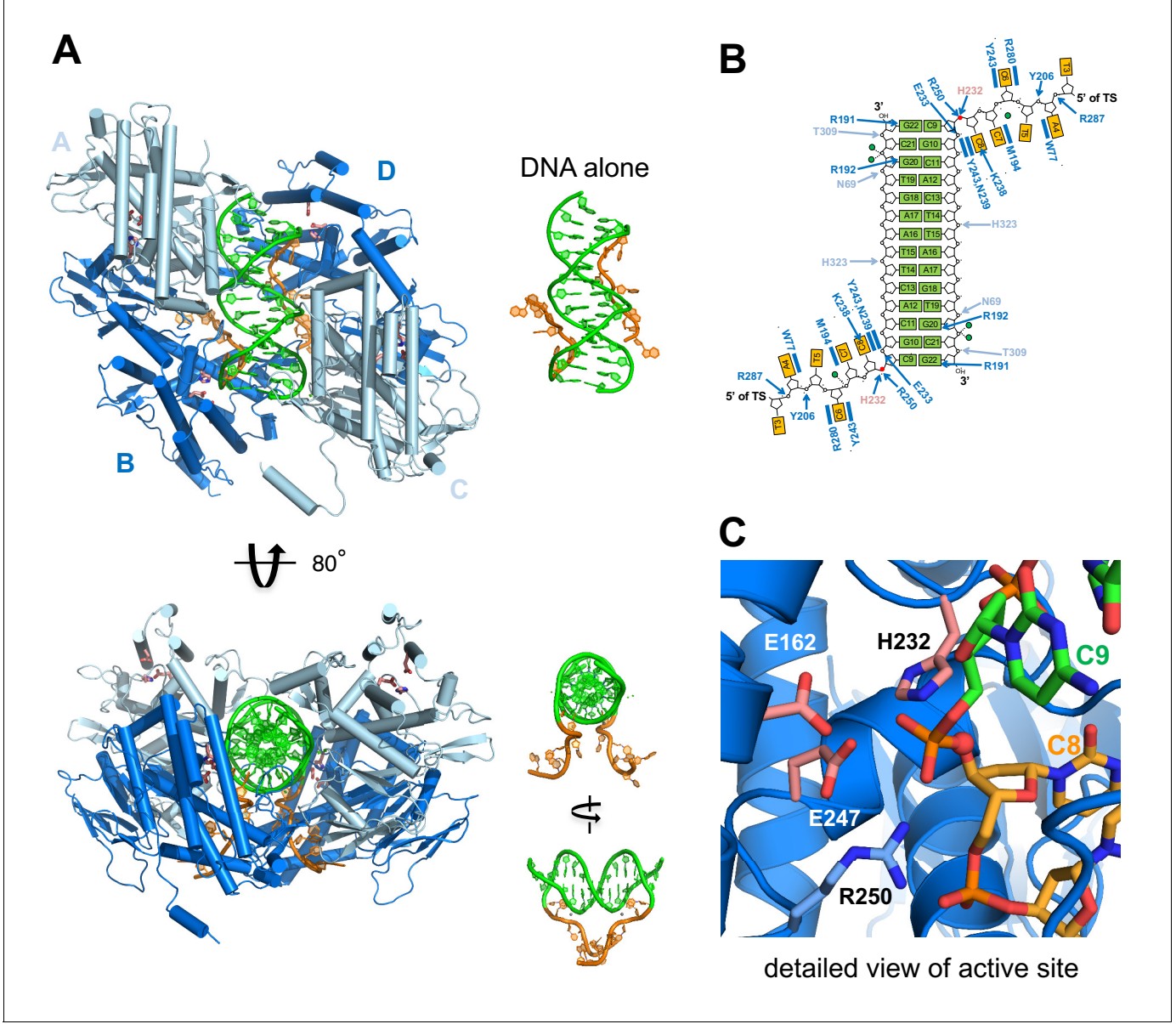

**Figure 5.** Structure of *M. mazei* casposase bound to an integration product mimic. (A) Two single-stranded casposon ends (5'-TATATCCC; shown in orange) are integrated into a 14 bp dsDNA target (5'-CGCACTTAAGCGTG; shown in green). This color code is maintained throughout. In the tetramer, catalytic subunits (**B and D**) are in dark blue and noncatalytic subunits in light blue (**A and C**). Views of the DNA alone are shown to the right of each view in the same orientation. Active site residues are shown as sticks (pink). (B) Schematic showing hydrogen bond contacts between protein and DNA using the same color code as (A). Red circles marks the phosphates at which integration occurred. Green circles correspond to bound calcium ions. (C) Detailed view of the active site.

The online version of this article includes the following figure supplement(s) for figure 5:

**Figure supplement 1.** Representative electron densities.

**Figure supplement 2.** Side-by-side views of the aligned structures of the casposase and Cas1 from *A. fulgidus* (PDB ID 4n06; *Kim et al., 2013*).

**Figure supplement 3.** Structure-based alignment of *M. mazei* casposase to *E. coli* (5vvk) and *E. faecalis* Cas1 (5xvp).

**Figure supplement 4.** SDS-PAGE analysis of dissolved crystals showing that the casposase was not proteolytically degraded.

## dsDNA target recognition

The *M. mazei* casposase generates 14 bp TSDs (*Figure 3A–C*), in sharp contrast to CRISPR-Cas systems where the CRISPR repeats that are the TSDs (generated after spacer integration and subsequent repair by DNA replication) are generally 25–50 bp. In the case of CRISPR-Cas integrases, both

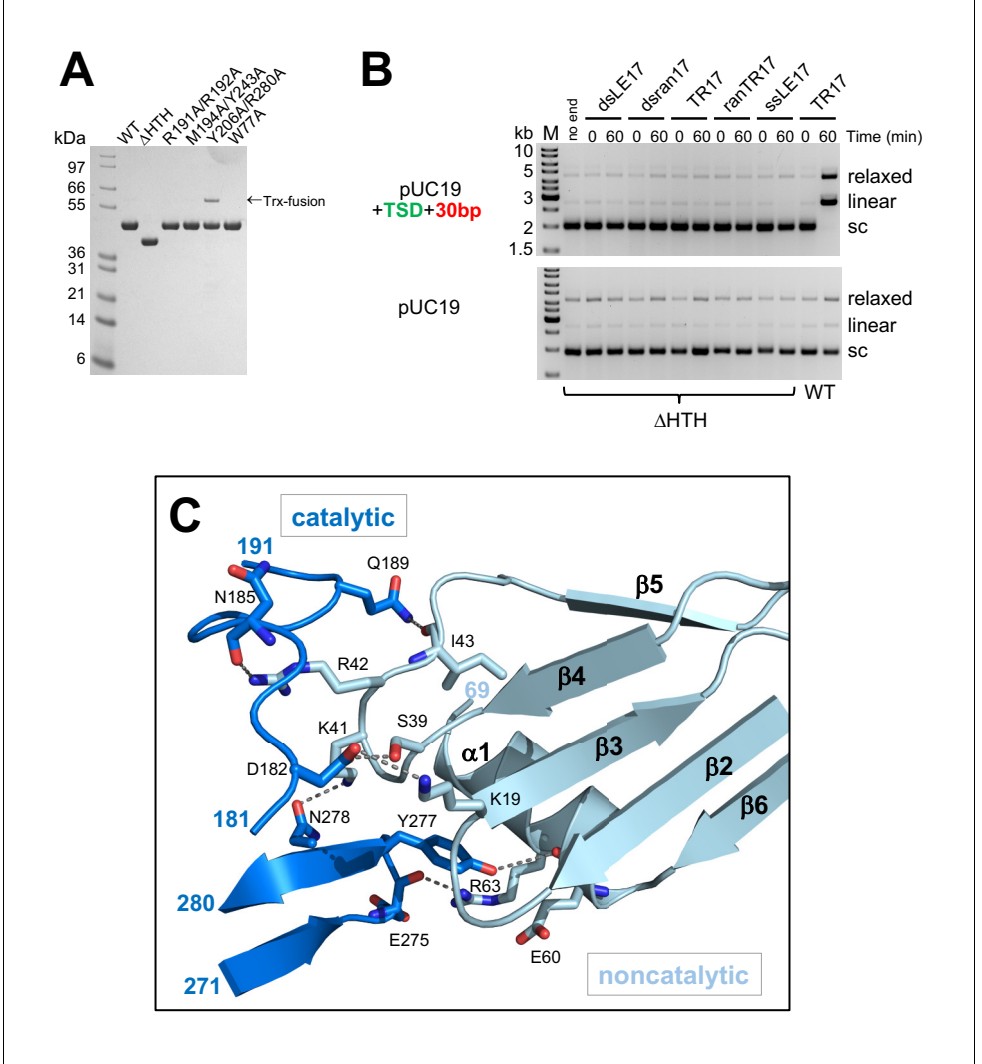

**Figure 6.** Details of the casposase structure. (**A**) SDS-PAGE analysis of proteins expressed and purified in this work. (**B**) Integration activity of the C-terminally truncated casposase, ΔHTH. There is no evidence for integration activity with either a target plasmid containing the casposon specific target site (top) or pUC19 alone (bottom). (**C**) Interactions at the casposase tetramerization interface. Shown are the hydrogen bond contacts between monomers A (noncatalytic) and D (catalytic), which are recapitulated between monomers B and C (not shown).

Cas1 and Cas2 participate in binding the CRISPR repeat (*Xiao et al., 2017*; *Wright et al., 2017*). In the casposase complex structure, the total casposase-dsDNA interface buries ~4700 Å$^2$ of solvent-accessible surface but only two specific contacts mediate the interaction with the target DNA: between R191 and G22 (i.e., the base opposite the first C of the 5′-CGCA motif) and between R192 and G20 (the base opposite the second C of the 5′-CGCA motif) (*Figure 7A,C*). The remaining inter-actions are non-specific, including contacts to the phosphate backbone from the main chain amides of T309 and E233 and from the side chains of N69 and H323. There is also a divalent ion-mediated interaction from a calcium ion contributed by the crystallization buffer with two phosphate groups and the carboxylate of D46. The bound target in the integration product complex is almost perfectly straight B-form DNA, unlike the severely bent targets observed in other DNA transpososomes and retroviral intasomes (for example, *Montaño et al., 2012*; *Maertens et al., 2010*).

To confirm the catalytic importance of R191 and R192, we expressed and purified a double mutant (R191A/R192A; *Figure 6A*) and tested its activity in the plasmid integration assay. When the optimal oligonucleotide substrate, TR17, was used, there was little or no relaxed or linear products observed (*Figure 7F*). We also tested a range of different casposon end substrates, and again there

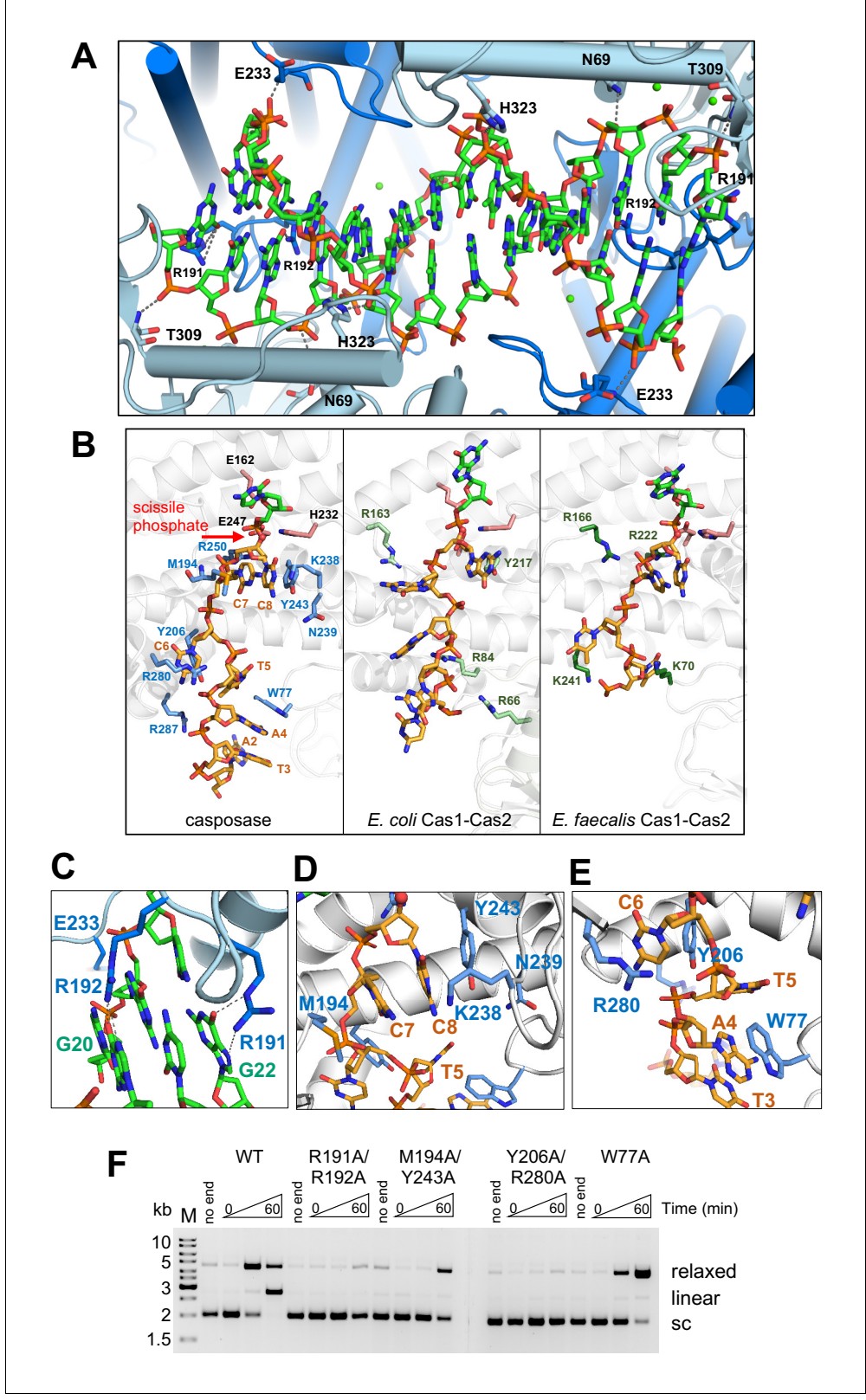

**Figure 7.** Binding of the casposon target site and casposon ends. (**A**) Hydrogen-bond contacts between the casposase and target DNA (casposon ends omitted for clarity). Bound Ca²⁺ ions are shown in green. (**B**) Comparison of binding of ssDNA casposon ends (left) and the ssDNA 3'-overhangs of bound spacers in the *E. coli* type I-E (center; 5vvk) and *E. faecalis* type II-A (right; 5xvp) integrases. Structures were aligned based on the

*Figure 7 continued on next page*

*Figure 7 continued*

catalytic monomers. Active site residues are shown in pink, and the red arrow indicates the 3'-OH group where the casposon end is joined to target. (C) Detailed view of the interactions of R191 and R192. (D) Detailed view of the interactions involving C7 and C8. (E) Detailed view of the interactions involving C6. (F) Integration activity of point mutants using pUC19+TSD+30 bp as the target plasmid and TR17 as the casposon end mimic. For each protein, time points are: 0, 1, and 60 min. Comparison of integration activity of purified casposase mutants with varying casposon end oligonucleotides is shown in *Figure 7—figure supplement 1*.

The online version of this article includes the following figure supplement(s) for figure 7:

**Figure supplement 1.** Comparison of integration activity of purified casposase mutants with casposon end oligonucleotides.

was little evidence for any activity (*Figure 7—figure supplement 1*), supporting the notion that these two arginines play important roles during integration.

## ssDNA casposon end binding resembles that of spacer 3′ overhangs

A rich set of protein-DNA contacts line the path of each single-stranded casposon end from the catalytic site to the protein surface (*Figure 7B*). Although the only base-specific interaction is between the first C of the casposon DNA (C8) and the side chain of K238, several residues interact with the phosphate backbone (H232, R250, Y206, R280, and R287) and there are arrays of stacking interactions. Most notably, the two first Cs of the casposon end are stacked on each other and both sandwiched between Y243 and M194; Y243 in turn is stacked on the side chain of N239 (*Figure 7D*). The third C of the casposon end (C6) is stacked between Y206 and R280 (*Figure 7E*). Finally, the fifth base of the end (A4) is stacked on W77 (*Figure 7E*). Through these extensive interactions, each single-stranded casposon end buries ~1700 Å total solvent-accessible surface area. We also note that the phosphate groups of C6 and C7 of each casposon end are located within 5 Å of the phosphate backbone of the target DNA, an unusually close approach stabilized in part by the binding of a $Ca^{2+}$ ion, which in turn is enabled by the syn conformation of C7.

To test the relative importance of the observed interactions in integration activity, we expressed and purified several mutants expected to disrupt sets of interactions including the double-mutants M194A/Y243A, Y206A/R280A, and the single mutant W77A (*Figure 6A*). Each of these mutants had severely reduced integration activity when assayed using TR17, most notably for the production of the linear product (*Figure 7F*). When other casposon end substrates were tested (*Figure 7—figure supplement 1*), there were varying levels of residual activity for the production of the relaxed form of the pUC19+TSD+30 bp plasmid (consistent with SE joining), but none of the mutant proteins generated a linear product consistent with double-ended integration. Interestingly, all of the mutants showed a higher activity level with a casposon-specific ssDNA end (ssLE17) than with non-specific DNA (ssran17).

The path followed by ssDNA from the casposase active sites is reminiscent of those of the 3′ overhangs of CRISPR spacers (*Wang et al., 2015*; *Nuñez et al., 2015b*; *Xiao et al., 2017*) although the number and type of interactions differ (*Figure 7B*). Consistent with the importance of ssDNA overhangs at the end of spacers, several CRISPR Cas1 proteins have been reported to bind ssDNA, and although ssDNA can be disintegrated by Cas1 (*Rollie et al., 2015*), stable half-site integration of single-stranded protospacers has not been detected (*Nuñez et al., 2015b*; *Fagerlund et al., 2017*; *Rollie et al., 2018*) although an exception has been reported (*Silas et al., 2016*). These collective observations suggest that the putative evolution from transposition to spacer integration may have involved the loss of the coupling between ssDNA binding and subsequent integration.

## Converting a casposase into a CRISPR-Cas integrase

To understand the structural relationship between the casposase integration product complex and CRISPR Cas1-Cas2 assemblies bound to DNA representing a spacer integrated into a CRISPR repeat, we first compared their aligned dimer structures. As can be seen in *Figure 8A*, the mode of target binding by the casposase resembles that of *E. faecalis* and *E. coli* Cas1 bound to CRISPR repeats, and the binding mode of the ssDNA casposon ends resembles that of 3′-overhangs of CRISPR spacers (*Wang et al., 2015*; *Nuñez et al., 2015b*; *Xiao et al., 2017*; *Wright et al., 2017*).

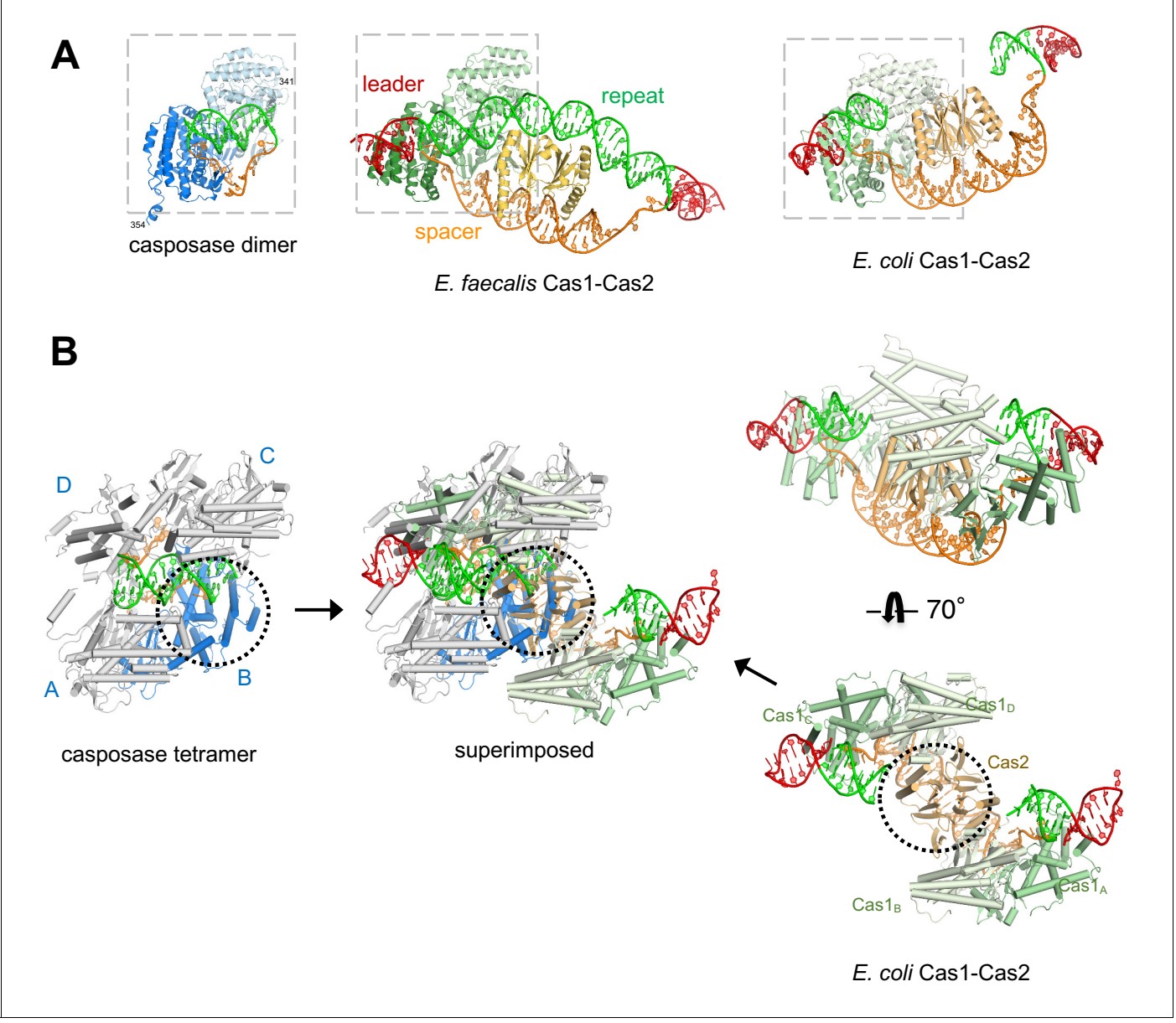

**Figure 8.** Features of the casposase-DNA complex. (A) Comparison of a DNA-bound casposon dimer (left) to full-site bound CRISPR Cas1-Cas2 integrases of *E. faecalis* type II-A CRISPR Cas1-Cas2 (PDB ID 5xvp, center) and *E. coli* type I-E (PDB ID 5vvk, right). Aligned dimers are boxed in gray. Casposon ends and spacers are shown in orange, target in green, and leader DNA in red. The final visible amino acid in the electron density on each casposase C-terminal end is indicated. (B) Superposition of the structures (center; casposase monomer D aligned to *E. coli* Cas1_C) shows that the location of the casposase monomer B (in blue) overlaps that of Cas2.

However, when the casposase tetramer is considered (*Figure 8B*), it is clear that one of the caspo-sase monomers occupies the same architectural space as one of the Cas2 subunits in CRISPR Cas1-Cas2 complexes (*Figure 8B*, center), and it would not be possible to retain the ability to tetramerize in the manner observed and to simultaneously bind Cas2.

At the level of the casposase dimer, however, there does not appear to be an obvious structural impediment to interacting with a Cas2 dimer (*Figure 9A*). Although the specifics of Cas1-Cas2 inter-actions are not conserved across CRISPR-Cas systems (*Wang et al., 2015*; *Nuñez et al., 2015b*; *Xiao et al., 2017*), when the relevant interfaces were compared, it appears that mutation of only a few surface residues of the casposase might be sufficient to convert it into one with both a comple-mentary shape and the necessary residues to bind Cas2. For example, the interaction between *E.*

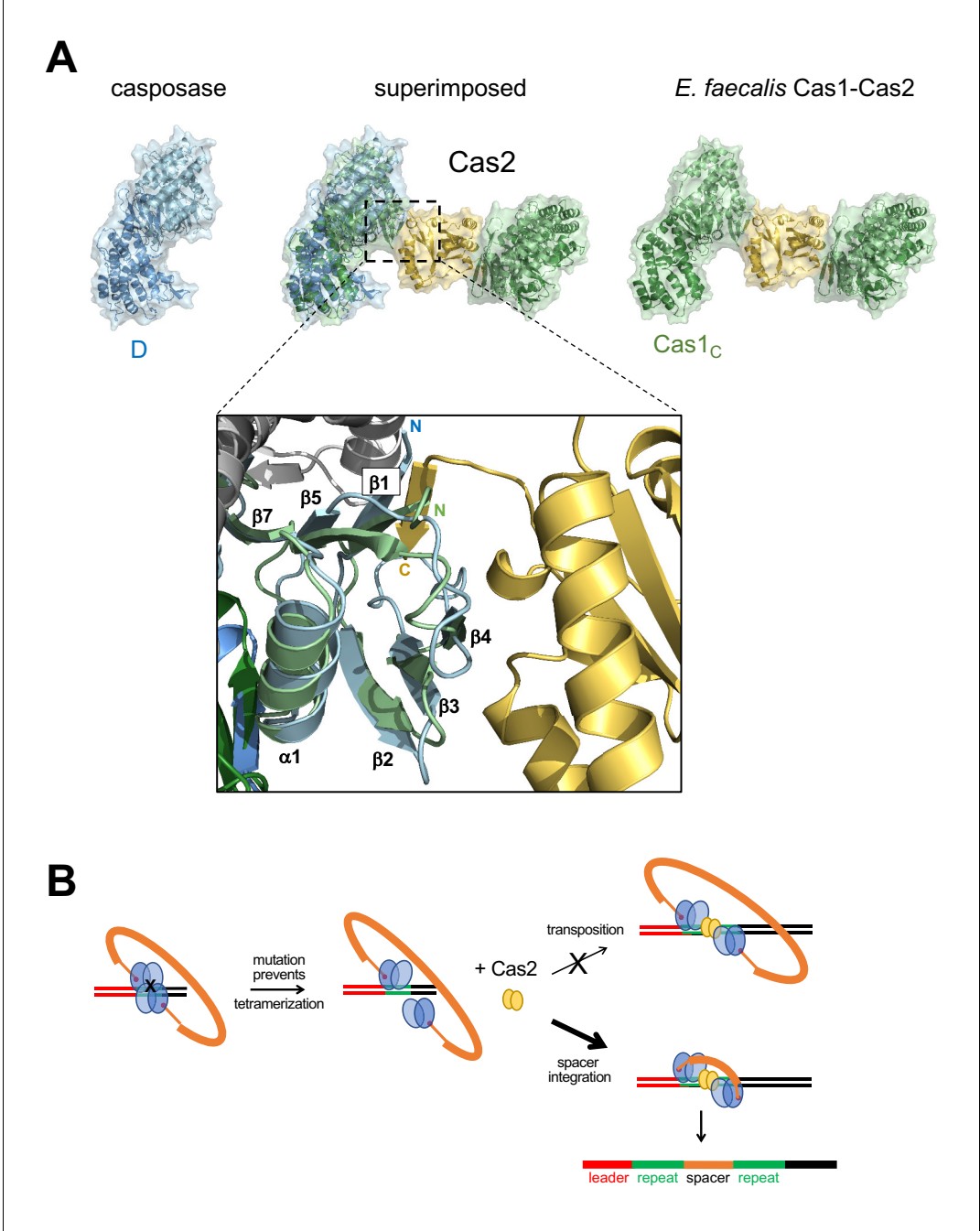

**Figure 9.** Proposed transformation from transposase to spacer integrase. (**A**) Comparison of the surfaces of casposase dimer and Cas1-Cas2 of *E. faecalis* (5xvp). The structures were aligned based on the catalytic monomers of each. Also shown is a close-up of the interaction between the *E. faecalis* Cas1 N-terminal domain (green) and Cas2 (gold); the aligned casposon N-terminal domain is overlaid in blue. In the inset, the view is based on the alignment between noncatalytic domains. (**B**) Proposed evolutionary transformation from casposase to CRISPR-Cas integrase in which Cas2 becomes a preferred binding partner of a casposase dimer, in turn leading to the preferential integration of a spacer over the original casposon. See *Figure 9—video 1*.

The online version of this article includes the following video for figure 9:

**Figure 9—video 1.** Possible structural evolution from a casposase tetramer to a Cas1-Cas2 heterohexamer.
https://elifesciences.org/articles/50004#fig9video1

*faecalis* Cas1 and Cas2 is mediated in part by a β-sheet formed between β1 of Cas1 and a β-strand at the C-terminus of Cas2, and β1 of the casposase appears similarly poised (***Figure 9A***).

It is not clear what changes in DNA recognition might need to accompany the architectural change from tetramer to Cas2-binder. Comparison of target binding sites in both integrase systems shows that the overall mode of binding is very similar, although the *M. mazei* casposase has base-specific contacts involving R191 and R192 that presumably would have to be lost since Cas1-Cas2 structures have shown little evidence for sequence-specific contacts with the CRISPR repeats (***Wright et al., 2017***; ***Xiao et al., 2017***). Another obvious difference is the lack of the casposase HTH domain in Cas1-Cas2, which we have shown is crucial for casposon integration. Until we have a better understanding of the role of the HTH, missing in the structure here, it remains unclear how Cas1-Cas2 compensated for its loss.

## Discussion

Evolutionary studies on CRISPR-Cas systems have suggested that some Cas proteins of effector complexes as well as Cas1 of the spacer integrase have arisen from mobile genetic elements (***Krupovic et al., 2014***; ***Koonin and Makarova, 2017***; ***Faure et al., 2019***). The deep branching of casposons within the Cas1 phylogeny (***Makarova et al., 2013***; ***Makarova et al., 2018***) and the clear mechanistic similarity between DNA transposition and CRISPR spacer acquisition makes the idea of an evolutionary relationship highly attractive. Like transposases and integrases with RNase H-like catalytic domains, the CRISPR-Cas integrase carries out transesterification to insert one piece of DNA into another, generates uniform-sized TSDs upon insertion, and produces a head-to-tail array of integrated DNA segments through tandem insertion as often seen with insertion sequences and transposons (***Siguier et al., 2015***). Yet, the putative evolutionary ancestor of the CRISPR-Cas integration system is not from one of the prevalent DNA transposon superfamilies but, rather, from the relatively sparsely distributed casposons (***Krupovic et al., 2014***). The reason for this may lie in the unusual capacity of the *M. mazei* casposase to integrate a variety of substrate types, including nonspecific and random DNA, into a strongly preferred target site, the important characteristic if a CRISPR array is to serve as an effective library of a range of previously encountered foreign DNA. Thus, on the hypothetical evolutionary pathway towards spacer integration, casposases appear well-suited to act as the starting point for a system to integrate a range of largely unrelated DNA sequences.

Our biochemical results described here indicate that the *M. mazei* casposase already possesses several observed properties of Cas1-Cas2 integrases (***Marraffini, 2015***; ***Sternberg et al., 2016***; ***Amitai and Sorek, 2016***; ***Jackson et al., 2017***), as has also been recently reported for the *A. boonei* casposase (***Béguin et al., 2019***). For example, the crucial identity of a few bp on each side of the top strand casposon integration site (***Figure 3E***, mutCGCA and mutL) corresponds to the conserved feature of CRISPR-Cas integration for specificity to the leader-repeat border. Specific integration of a transposon avoids any possible deleterious effects of random integration on the host cell genome, whereas for CRISPR-Cas, this also ensures that the latest encountered cellular invader is effectively neutralized as it generates the most robust immune response (***McGinn and Marraffini, 2016***). Furthermore, the requirement for a sequence motif upstream of the casposon integration site (***Figures 1B*** and ***3E*** mut12-17) appears to correspond to the observation that in some CRISPR-Cas systems, sequences upstream within the leader are important for integration (***Alkhnbashi et al., 2016***). This requirement is understood for the *E. coli* Type I-E system in which IHF binds within the last 40 bp of the leader and is needed to bend and deform the target site so that integration can occur (***Nuñez et al., 2016***; ***Wright et al., 2017***). It is possible that the role of the casposase HTH domain is to recognize a specific sequence just upstream of the integration site and thereby direct integration; this would be consistent with our observation that it is crucial for activity (***Figure 6B***). We have expressed and purified the HTH domain alone, but have been unable to confirm an interaction with DNA (data not shown). As these shared properties are already features of casposon transposition, they are not a consequence of the evolution of one system into the other.

Based on the biochemistry and structure presented here, we suggest an evolutionary pathway in which the crucial change was architectural (***Figure 9B***, ***Figure 9—video 1***). In this scenario, the ability of the casposase to tetramerize on DNA upon target binding was lost when it acquired a preference for binding to a Cas2 molecule rather than to itself. This seems a reasonable possibility as the total protein surface buried by the casposase by tetramerization (~2400 Å$^2$) is substantially smaller

than the total Cas1-Cas2 buried surface in the heterohexamer of either *E. coli* (~7800 Å$^2$; *Nuñez et al., 2014*) or of *E. faecalis* (~4000 Å$^2$; *Xiao et al., 2017*). As shown by the structure here, if an intervening Cas2 dimer were introduced as a bridge between the two noncatalytic subunits of the casposase tetramer, the active sites of the catalytic monomers would not only remain oriented towards each other but Cas2 would provide an additional DNA binding surface that could preferentially direct a single, short piece of DNA between two active sites.

In this model, the CRISPR repeat sequence would be therefore derived from the original target site preferred by the casposase, and any original specificity for the casposon terminal inverted repeats would devolve to a residual spacer sequence preference. It is also possible that one evolutionary path could conserve the ability of the *M. mazei* casposase to integrate RNA as well as DNA (*Silas et al., 2016*). We note that the RAG1 protein of the V(D)J recombinase, also derived from a DNA transposon, conserves properties of its original transposase in that specific signal sequences are recognized and cleaved out from its flanking coding sequences (*Schatz and Swanson, 2011*) while the random integration activity has been severely reduced to avoid genomic damage. Thus, in both cases, it is the logic of the particular system (i.e., biochemical properties of its transposase) that has been evolutionarily conserved.

In the evolutionary model presented here, when Cas2 was acquired as a preferred binding partner, it seems likely that Cas1 active sites would be forced farther apart in CRISPR integrases than in casposases, thereby expanding the length of the target site duplication. As has been pointed out (*Wright et al., 2019*), this could have been important to allow the sampling of longer pieces of DNA which would produce more specific crRNA for incorporation into effector complexes at a later step in the immunity process. However, it is not yet clear why - or even if - the two integration pathways are necessarily mutually exclusive, if appropriate substrates are supplied. Indeed, the presence in *M. mazei* strains of 'solo-TIRs' of specific length (either 19 or 31 bp; *Krupovic et al., 2016*) might be evidence of the capacity of a casposase to integrate short spacer-like sequences or an early step along the path to the development of a CRISPR array (*Jackson et al., 2017*). It is an intriguing possibility that some casposases may already possess the ability to integrate short CRISPR-like spacers, and nothing in the casposase structure appears to rule it out except perhaps the tight quarters of the compact tetramer and the lack of an extended dsDNA binding surface such as that provided by Cas2.

It has recently been reported that a type V-C Cas1 protein is active in vitro for spacer integration in the absence of Cas2, and has been proposed to be an early ancestor of the Cas1-Cas2 spacer acquisition system (*Wright et al., 2019*). Like the casposase here, the V-C Cas1 forms a tetrameric assembly upon binding to oligonucleotides that represent spacer integration products but differs in three key features. The first is its exquisite preference for spacers that are recessed by 0–2 nt on one strand: spacers that are recessed by three nt or longer are not functional for spacer integration whereas the *M. mazei* casposase is capable of integrating blunt-ended substrates, substrates with varying numbers of single-stranded nucleotides at its 3'-end, as well as completely single-stranded casposon ends of varying length. The second difference from casposases is the length of the target site as the 25 bp V-C target site, although shorter than average for CRISPR-Cas systems, is substantially larger than the ~14 bp average for casposases (*Krupovic et al., 2014*). Finally, the V-C Cas1 is less sequence-specific in its integration site preference relative to other characterized Cas1-Cas2 integrases (*Wright et al., 2019*), in marked contrast to the stringent sequence-specificity we observed here for the *M. mazei* casposase (*Figure 3B*). It will be extremely interesting to understand the structural basis of the transformation from casposase to a functional tetrameric spacer integrase.

The structure here provides insight into how a casposase recognizes single-stranded casposon ends and its target site sequence, but the target DNA used does not contain all of the elements required for site-specific integration. Determining the role of sequences just upstream of the top strand integration site may provide insight into the ancestral role of what became the critical CRISPR leader-repeat border. Another intriguing question relates to the role of the casposase C-terminal domain. It is tempting to speculate that it might play a role similar to that of IHF in the Type I-E system in bending target (*Nuñez et al., 2016*), perhaps through recognition of a specific sequence upstream of the integration site, but the answer awaits further investigation. Our results explain the capability of casposases to use ssDNA as an integration substrate but do not shed light on the nature of the in vivo substrate. As it is not known if an integratable casposon is generated through an excision mechanism or through the process of replication, where ssDNA fits into the puzzle

remains to be established. The only other transposons known to site-specifically integrate ssDNA are those of the IS200/IS605 family (**Barabas et al., 2008**). However, their integration is fundamentally different as both the 3' and 5' ends of the elements are joined to a single-stranded target. It will be interesting to determine if the effectiveness of ssDNA casposon ends for integration might be linked more closely to the mechanism of casposon mobility, as originally proposed (**Krupovic et al., 2014**). Certainly, the fellow-traveling *polB* gene in casposons is suggestive that ssDNA and replication may be central to casposon mobility.

# Materials and methods

### Key resources table

| Reagent type (species) or resource | Designation | Source or reference | Identifiers | Additional information |
|---|---|---|---|---|
| Gene (Methanosarsina mazei) | casposase | | WP_011035139.1 | |
| Strain (*E. coli*) | Top10 | Invitrogen/ ThermoFisher | C404003 | |
| Strain (*E. coli*) | B834(DE3) | MilliporeSigma | 69041–3 | |
| Commercial kit | Selenomethionine Medium Complete | Molecular Dimensions | MD12-500 | |
| Recombinant DNA reagent | pUC19 | Invitrogen | P/N 54357 | |
| Chemical compound, drug | PEG 8000 | Hampton Research | HR2-535 | |
| Chemical compound, drug | CHES, pH 9.0 | Hampton Research | HR2-256-05 | |
| Commercial kit | Nextera XT DNA sample prep | Illumina | #FC1311024 | |
| Software, algorithm | *Rsubread* package | http://bioconductor.org/ packages/Rsubread | V1.34.7 | DOI: 10.18129/ B9.bioc.Rsubread |
| Software, algorithm | Gviz package | https://bioconductor.org/ packages/Gviz/ | | V1.26.5 |
| Software, algorithm | *ggseqlogo* | https://www.rdocumentation.org/ packages/ ggseqlogo/versions/0.1 | | V0.1 |
| Software, algorithm | *Biostrings* | https://bioconductor.org/packages/ release/bioc/html/Biostrings.html | | V2.50.2, RRID:SCR_016949 |
| Software, algorithm | *Rsamtools* | | | V1.34.1 |
| Software, algorithm | *ShortRead* | | | V1.40.0 |
| Software, algorithm | *GenomicAlignments* | | | V1.18.1 |
| Software, algorithm | *Jalview* | www.jalview.org | | V1.0, RRID:SCR_006459 |
| Software, algorithm | XDS | xds.mpimf-heidelberg.mpg.de | | RRID:SCR_015652 |
| Software, algorithm | XPREP | https://www.bruker.com | Bruker | |
| Software, algorithm | SHELXD | http://shelx.uni-ac.gwdg.de/ | | **Usón and Sheldrick, 1999**; RRID:SCR_014220 |
| Software, algorithm | SHARP | https://www.globalphasing .com > sharp | | **Bricogne et al., 2003** |
| Software, algorithm | CCP4 | https://www.ccp4.ac.uk | | **Winn et al., 2011**; RRID:SCR_007255 |
| Software, algorithm | DM | https://www.ccp4.ac.uk | | **Cowtan, 1994**; RRID:SCR_007255 |
| Software, algorithm | O | http://xray.bmc.uu.se/ alwyn/TAJ/Home.html | | **Jones and Kjeldgaard, 1997** |
| Software, algorithm | CNS1.3 | http://cns-online.org/v1.3/ | | **Brunger, 2007** |

*Continued on next page*

*Continued*

| Reagent type (species) or resource | Designation | Source or reference | Identifiers | Additional information |
|---|---|---|---|---|
| Software, algorithm | BUSTER 2.10.3 | https://www.globalphasing.com > buster | | *Bricogne et al., 2017* RRID:SCR_015653 |
| Software, algorithm | Phenix 1.10 | www.phenix-online.org | | *Adams et al., 2010* RRID:SCR_014224 |
| Software, algorithm | PyMOL v1.7 | https://pymol.org | Schrodinger, Inc | RRID:SCR_000305 |

## Cloning, protein expression, and purification

The gene encoding the casposase from *Methanosarsina mazei* strain 3.F.A.1A.1 (WP_011035139.1) was cloned into a modified pBAD vector as previously described for the *A. boonei* casposase (*Hickman and Dyda, 2015*). For optimal expression and purification, the single point mutation C184S was introduced and 17 additional amino acids were added between the N-terminal TEV protease cleavage site and M1 to allow complete removal of the thioredoxin (Trx) tag. Purification was essentially as previously described (*Hickman and Dyda, 2015*). For selenomethionine-substituted protein, the coding sequence corresponding to the Trx-casposase fusion protein was subcloned into pET-29b, and transformed into *E. coli* strain B834(DE3). Expression was performed using the M9 minimal media kit and selenomethionine from Molecular Dimensions Inc (Maumee, OH) according to the manufacturer's instructions. Full selenomethionine substitution was confirmed by LC/MS (Biochemistry Core Facility, NHLBI). Purification and crystallization were identical to those of the unsubstituted protein. All purification steps were carried out at 4°C.

A C-terminally truncated casposase, residues 1-341 (ΔHTH), was generated by the deletion of residues 342-406 from the Trx-casposase fusion protein by PCR mutagenesis. Expression and purification was identical to that of the full-length version. Single and double point mutations were introduced by PCR, and sequences of the full coding frame confirmed. Purification was essentially as the wild-type with the exception of the Y206A/R280A double mutant, which yielded only ~10% of the level of soluble protein when compared to wild-type and was not completely cleaved by TEV, despite prolonged incubation. Due to the low level of recovered protein, it was assayed without performing the usual final size exclusion chromatography step.

## Enzymatic assays

### In vitro DNA integration into pUC19 and modified pUC19 target plasmids

Oligonucleotides (IDT) were resuspended in 10 mM Tris and, as needed, mixed in the appropriate molar ratio, heated to 95°C for 15 minutes, and allowed to slowly cool to room temperature. Modified pUC19 target plasmids were constructed using the QuikChange method by introducing target sequences between bp 575 and 576. For assays, purified protein (75 nM) and oligonucleotide (200 nM) were mixed in buffer consisting of 25 mM Tris pH 7.5, 150 mM KCl, 0.05 mg/ml BSA, and 5 mM $MnCl_2$ and incubated at room temperature for the times indicated. For each set of reactions, a "no end" control was performed in which the oligonucleotide was omitted. Reactions were initiated by the addition of 150 ng target plasmid and stopped at time points indicated by adding EDTA to a final concentration of 25 mM followed by proteinase K digestion for 30 min at 37°C (NEB; 40 U per reaction tube). Glycogen was then added, the reaction products ethanol-precipitated, and the tubes air-dried. Products were resuspended in 1X DNA loading buffer, run on a 1.5% agarose gel in 1X TAE at 100V, and visualized using ethidium bromide. Bands were quantified where indicated using ImageJ.

### In vitro DNA integration into oligonucleotide targets

Protein (1.4 μM) and casposon end oligonucleotides (1.2 μM) were mixed in buffer consisting of 25 mM Tris pH 7.5, 150 mM KCl, 0.05 mg/ml BSA, and 5 mM $MnCl_2$. Reactions at room temperature were initiated by the addition of the target oligonucleotide, and were stopped at the time points indicated by mixing 1:1 with formamide gel buffer. Reactions were run on 15% acrylamide TBE-urea gels (Novex) at 250V in 1X TBE and visualized using SYBR Gold (Invitrogen). Bands were quantified as indicated using the AlphaView band analysis software of an Alpha Innotech imager.

## Library preparation and sequencing

To prepare four of the DNA samples for sequencing, the in vitro integration reaction was carried out in quadruplicate for 60 min using dsLE31 and pUC19+TSD+30bp as described above. The resulting samples were run on 1.5% agarose and, for each, the band corresponding to the linear product of integration was isolated using the QIAquick Gel Extraction Kit (Qiagen). The resulting DNA was then used as the template for four independent PCR reactions using a LE31 primer to amplify the product. The reaction was carried out for either 17 or 30 cycles. Amplified products corresponding to the size of linearized pUC19 were again isolated from agarose prior to library preparation. For the fifth sample, the integration reaction was scaled up four-fold and carried out in triplicate for 2 hours using TR29 as the integration substrate. Bands corresponding to the linear integration product were isolated from agarose, subjected to one round of extension using Pfu Ultra II Fusion DNA polymerase (Agilent) for 30 sec at 72°C, and purified again using the QIAquick PCR purification kit (Qiagen). The five sequencing libraries were prepared from the supplied DNA using Nextera XT DNA sample prep kit (Ilumina, #FC1311024) according to the manufacturer's protocols. The libraries were then sequenced in a stranded 75 bases paired-end run on the MiSeq instrument and analyzed.

## NGS computational analysis

The read 1 fastq files for each sample were processed, aligned and analyzed in *R* (V3.5.1) using custom scripts. Specifically, the reads were first collapsed by sequence, abundance of individual sequences was recorded, and results were stored in fasta format. Each fasta file was filtered, extracting only the unique sequences containing casposon end sequence (5'-GAGTTACCTATATCCC) for further analysis. These sequences were then trimmed, removing the casposon end fragment and retaining only downstream regions for aligned to the plasmid using *Rsubread* package (V1.34.7). Generated alignment files were used to visualize sequence alignments and coverages using *Gviz* package (V1.26.5). To generate sequence logos, the aligned sequences were first extracted from alignment files by strand, and each sequence was then duplicated based on the recorded abundance information obtained from the fastq files. The sequences were then aligned at the 5'end and alignment was used to calculate nucleotide frequency matrix extending from positions +1 up to +30 nucleotides downstream of integration of casposon end sequence. The logos were then visualized using *ggseqlogo* (V0.1) and negative strand logos flipped using Adobe Illustrator to reflect the complimentary base pairing. The alignment, conservation and upstream sequences of different target sites were performed and depicted using *Jalview* (V1.0) software. Additional packages used in the analysis include *Biostrings* (V2.50.2), *Rsamtools* (V1.34.1), *ShortRead* (V1.40.0), *GenomicAlignments* (V1.18.1). The scripts are available upon request. The fastq files have been deposited to GEO under the accession number: GSE139037.

## Assay of DNA binding by size exclusion chromatography

After samples were mixed at various protein:DNA ratios and dialyzed overnight into 35 mM CHES pH 9.0, 0.5 M KCl, 105 (w/v) glycerol, 0.2 mM TCEP, protein multimerization and DNA binding was assessed. Samples were injected onto a Superdex 200 Increase 3.2/300 column equilibrated in dialysis buffer and eluted at 0.05 ml/min at 4°C. A standard curve was generated using solutions at 1 mg/ml in dialysis buffer of blue dextran, thyroglobulin (669 kDa), apoferritin (443 kDa), β-amylase (200 kDa), alcohol dehydrogenase (150 kDa), bovine serum albumin (66 kDa), and carbonic anhydrase (29 kDa).

## Interferometric scattering mass spectrometry (iSCAMS)

Samples at ~1 mg/ml (~25 μM) were diluted to 25-50 nM in buffer containing 30 mM CHES pH 9.0, 2 mM EDTA, and either 1 M or 0.5 M KCl immediately prior to measurement using a G10 RefeynOne mass photometer. For each experiment, thousands of protein molecules were counted and their contrasts measured. Contrast values were converted to molar masses using a standard curve prepared using BSA (monomer+dimer), alcohol dehydrogenase (monomer+dimer), ovalbumin, and thyroglobulin in PBS buffer. Distributions were fitted using a Gaussian distribution model.

## Complex formation for crystallography

Oligonucleotides were combined in several molar excess with purified casposase and dialyzed overnight into Complex Buffer 1 (30 mM CHES pH 9.0, 0.5 M KCl, 10% w/v glycerol, 0.2 mM TCEP, 0.2 mM EDTA). This was followed by size exclusion chromatography in the same buffer on a 16/60 Superdex 200 column. Fractions containing the complex were subsequently combined and concentrated at room temperature in ThermoScientific 30K CO units. Glycerol was removed by extensive dialysis against Complex Buffer 1 without glycerol. Complexes were stored at 10°C to prevent cold-induced precipitation.

## Crystallization and structure determination

Crystallization was performed in hanging drop format at room temperature. Briefly, protein-DNA complexes were mixed with equal volume of reservoir solution containing 0.15-0.185 M calcium acetate, 84 mM sodium acetate pH 4.4-3.8, 7.5-9% PEG 8000, and 0.5 M KCl. Crystals grew to final size within 48h, and addition of 0.5 M NaCl to the reservoir encouraged further growth. Crystals were cryoprotected by transfer to mother liquor and quick serial transfer in increasing concentrations of ethylene glycol to 15-17.5% (v/v) final. Data was collected at the Southeast Regional Collective Access Team (SERCAT) beamline ID22 of the Advanced Photon Source.

Diffraction data collected at three energies around the Se K edge (*Table 1*) were integrated and scaled with XDS. Zero-dose (*Diederichs et al., 2003*) corrected data were used to calculate Fa coefficients (absolute value of the energy independent part of the Se structure factors) in XPREP (Bruker) and 27 Se positions were found with SHELXD (*Usón and Sheldrick, 1999*). The positional and occupancy parameters of the Se substructure were optimized with SHARP (*Bricogne et al., 2003*) and additional 5 Se sites were added. Solomon solvent-flattened electron density map was used to generate masks for non-crystallographic symmetry (NCS) averaging using CCP4 (*Winn et al., 2011*). The asymmetric unit contained a tetramer of the casposase composed of two dimers related by a proper two-fold axis. However, the monomers forming the dimers were not related by a proper rotational symmetry. Therefore, one NCS averaging mask was generated for a catalytic domain and another for one of the dimers. Multi-domain non-crystallographic symmetry averaging, further solvent flattening, histogram matching, and slow phase extension were carried using DM (*Cowtan, 1994*).

The resulting experimental electron density map was of excellent quality at 3.2 Å with clearly visible side chain densities at most places and clear density for the bound DNA. The molecular model was built using O (*Jones and Kjeldgaard, 1997*) and refined with several cycles of cartesian molecular dynamics, energy minimization and ADP refinement using CNS1.3 (*Brunger, 2007*) while the final refinement was carried out with BUSTER 2.10.3 (*Bricogne et al., 2017*). Composite simulated annealed omit map based on the final refined model for model verification was computed using Phenix 1.10 (*Adams et al., 2010*). The current model contains all residues from M1 to I341 in all four protein chains and to L354 for the B chain and V359 in the C chain; however, due to the lack of interpretable side chain densities, the last few residues in the B and C chains were modeled as serine. There was no visible connected electron density for the C-terminal domain; however, some disconnected but clearly helical densities were observed. These were modeled as polyserines.

Interfaces were analyzed using Protein interfaces, surfaces and assemblies' service PISA (*Krissinel and Henrick, 2007*) at the European bioinformatics Institute (http://www.ebi.ac.uk/pdbe/prot_int/pistart.html). All structural figures were generated using PyMOL (Schrodinger, Inc).

## Acknowledgements

We thank Dalibor Kosek for assistance with X-ray data collection, Duck-Yeon Lee of the Biochemical Core Facility at the NHLBI for LC/MS support, and Greg Piszczek and Di Wu of the Biophysics Core Facility at the NHLBI for help with iSCAMS measurements. Next-generation sequencing was performed by the NHLBI Sequencing and Genomics Core. This work was supported by the Intramural Program of the National Institute of Diabetes and Digestive and Kidney Diseases, National Institutes of Health. Data were collected at Southeast Regional Collaborative Access Team (SER-CAT) 22-ID beamline at the Advanced Photon Source, Argonne National Laboratory. SER-CAT is supported by its member institutions (www.ser-cat.org/members.html), and equipment grants (S10_RR25528 and S10_RR028976) from the National Institutes of Health. Use of the Advanced Photon Source was

supported by the U. S. Department of Energy, Office of Science, Office of Basic Energy Sciences, under Contract No. W-31–109-Eng-38. Some of the computations were carried out using the High Performance Computing Systems at the NIH.

## Additional information

### Funding

| Funder | Grant reference number | Author |
| --- | --- | --- |
| National Institute of Diabetes and Digestive and Kidney Diseases | Intramural Program | Alison B Hickman<br>Shweta Kailasan<br>Pavol Genzor<br>Astrid D Haase<br>Fred Dyda |

The funders had no role in study design, data collection and interpretation, or the decision to submit the work for publication.

### Author contributions

Alison B Hickman, Conceptualization, Formal analysis, Supervision, Validation, Investigation, Visualization, Methodology; Shweta Kailasan, Conceptualization, Formal analysis, Validation, Investigation, Methodology; Pavol Genzor, Astrid D Haase, Conceptualization, Formal analysis, Methodology, Writing - review and editing; Fred Dyda, Conceptualization, Resources, Formal analysis, Supervision, Funding acquisition, Validation, Investigation, Visualization, Methodology

### Author ORCIDs

Alison B Hickman (iD) http://orcid.org/0000-0001-7666-0249
Shweta Kailasan (iD) http://orcid.org/0000-0003-0876-6812
Fred Dyda (iD) https://orcid.org/0000-0003-1689-9041

### Decision letter and Author response

Decision letter https://doi.org/10.7554/eLife.50004.sa1
Author response https://doi.org/10.7554/eLife.50004.sa2

## Additional files

### Supplementary files

• Supplementary file 1. DNA oligonucleotides used. Random oligonucleotide sequences were generated using http://www.faculty.ucr.edu/~mmaduro/random.htm and were rejected only if they included a 3'-C nucleotide. Red indicates change from targ40 sequence.

• Transparent reporting form

### Data availability

Diffraction data for the casposase-DNA complex have been deposited under PBD ID 6OPM. The GSE number for the NGS data is GSE139037.

The following datasets were generated:

| Author(s) | Year | Dataset title | Dataset URL | Database and Identifier |
| --- | --- | --- | --- | --- |
| Hickman AB, Kailasan S, Genzor P, Haase AD, Dyda F | 2019 | Casposase structure and the mechanistic link between DNA transposition and spacer acquisition by CRISPR-Cas | https://www.ncbi.nlm.nih.gov/geo/query/acc.cgi?acc=GSE139037 | NCBI Gene Expression Omnibus, GSE139037 |
| Hickman AB, Kailasan S, Genzor P, Haase AD, Dyda F | 2019 | NGS data | https://www.rcsb.org/structure/6OPM | RCSB Protein Data Bank, 6OPM |

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
