## [Decision Letter]

**Acceptance summary:**

Cas1 and Cas2 are characteristic of the CRISPR-Cas adaptive immune systems found in most archaeal and many bacterial genomes. It is hypothesized that Cas1 evolved from a rare class of transposases, called casposases. Here, Hickman et al. express and purify a casposase from *Methanosarsina mazei*. They demonstrate that the MmCasposase integrates a wide variety of RNA and DNA substrates into a specific location. Integration requires substrates with a free 3'-OH and a target that contains specific sequence motifs. To understand the mechanism of integration the authors crystallize the casposase bound to DNA. The structure reveals a tetrameric architecture in which two casposase dimers sandwich the target DNA. The authors compare this structure to available crystal structures of Cas1 and Cas1-2 integration complexes. This analysis shows that the casposase tetramer is mutually exclusive with Cas2 binding and suggests how architectural changes in the Cas1-Cas2 complex may have led to mechanistic differences between the two types of integrases.

**Decision letter after peer review:**

[Editors’ note: the authors submitted for reconsideration following the decision after peer review. What follows is the decision letter after the first round of review.]

Thank you for submitting your work entitled "Casposase structure and the mechanistic link between DNA transposition and spacer acquisition by CRISPR-Cas" for consideration by *eLife*. Your article has been reviewed by three peer reviewers, one of whom is a member of our Board of Reviewing Editors, and the evaluation has been overseen by a Senior Editor. The reviewers have opted to remain anonymous.

Our decision has been reached after consultation between the reviewers. Based on these discussions and the individual reviews below, we regret to inform you that your work will not be considered further for publication in *eLife*, in its present form.

In this manuscript, Hickman et al. present the first crystal structure of a transposon-encoded casposase bound to an integration product mimic. The structure reveals a tetrameric architecture in which two casposase dimers sandwich the target DNA. The authors compare this structure to available crystal structures of Cas1 and Cas1-2 complexes. This analysis shows that the casposase tetramer is mutually exclusive with Cas2 binding and suggests how architectural changes in the Cas1-Cas2 complex may have led to mechanistic differences between the two types of integrases.

The reviewers all agree that the structure presented in this manuscript is potentially important. However, the structural data are not supported by biochemical experiments that advance our understanding of mechanism.

Some of the specific points raised during the review and in the discussion between reviewers and the editors are noted below, and the individual reviews are appended. We concluded that it could take substantial time to address all of these issues in a satisfactory manner. At this point, the authors may simply wish to submit this high quality paper elsewhere, without delay. If, however, these points can be addressed satisfactorily then the paper could be resubmitted to *eLife* for further consideration, should the authors wish to do so.

1) Clarification on the DNA substrates used in the biochemical assay, and an explanation for how these substrates relate to the biology are required. Additional controls and additional time points are necessary. Important controls are missing from the results presented in Figure 1, most notably negative controls in which the plasmid is incubated with casposase in the presence of divalent metal ions but no transposon end substrates. Previous Cas1 studies have observed considerable nicking activity (or possibly nicking/cleavage from contaminating proteins), so it is crucial that the baseline activity of the casposase protein prep on plasmid DNA is evaluated. Furthermore, for the data presented in panels C and D, no lanes exist showing the starting plasmid substrate at time 0 (i.e. in the absence of protein and transposon end substrate); since any given plasmid prep will have variable ratios of supercoiled to open-circular plasmid, it is critical that the reader have this comparison – for the exact substrate used in the experiments – to fully evaluate the biochemical activity of casposase and discriminate this from the starting status of the substrate. Labeling the bands "target," "SE," and "DE," is misleading, since a plasmid will naturally purify from a bacterial culture with some fraction in the open circular form, which will migrate the same as a SE product but is certainly not a single-end integration product.

2) Provide more experimental support and/or justification for the specific choice of the 8-nt overhang integration product. Why was 8 selected as the overhang length? Was a larger crystallization screen performed, and this was the only substrate geometry that resulted in diffracting crystals? Is there any further evidence that this is a physiologically relevant ssDNA end sequence? For at least one reviewer, it was unclear what the SEC data provided that was not show by the iSCAMS analysis. All reviewers commented on misplaced labels and inappropriate units (time rather than volume) in the SEC. There was some discussion about moving the SEC to the supplement but regardless of where it is discussed, the interpretation of these data must be clarified.

3) The authors gloss over the interesting finding that the HTH domain is disordered in the crystals and fail to follow-up with further experiments to investigate the role of this domain on biochemical integration activity, and whether the domain may be important in the context of more physiologically relevant substrates.

4) A major take-away from this manuscript would be that this casposase integrates substrates primarily at a specific target site rather than randomly, as has been previously touched upon in the literature. However, apparent integration events in pUC19 alone and farther away from the target site have been detected in this and previous studies. No sequencing data for these events are shown, and the authors fail to perform additional analysis on integration patterns. The sequences should be shown and discuss. It has become routine in spacer acquisition studies with Cas1 to deep sequence half-site integration products in order to gain deeper insights into specificity. Similar NGS experiments are warranted here to enhance their biochemical results. While NGS experiments are not required for publication in *eLife*, the authors should perform bioinforamtics analyses to determine whether related casposons are always present in genomes adjacent to similar target sites. Please mention, in the Discussion, the recent observation that the Type V-C Cas1 is much less sequence-specific in its integration site preference.

5) When describing the structure, the authors note several protein-DNA contacts, but it is unclear whether any of these contacts contribute to the specificity for the TSD target or LE substrate. From the data shown in Figure 1C, it appears that the casposase is somewhat specific for the LE sequence, especially for shorter single-stranded DNAs. Can this specificity be explained by the structure? Do R191 and R192 contribute to the specificity of the casposase for the TSD? The authors speculate about the importance of stacking interactions for ssDNA substrate binding (subsection “ssDNA casposon end binding resembles that of spacer 3' overhangs”). Did the authors perform any mutagenesis experiments to establish the importance of these contacts for target and substrate specificity? Experiments supporting structure-based hypotheses proposed by the authors are required.

6) The order in which substrates are shown on the left gel in Figure 1C is confusing. If possible, it would be helpful if this gel could be re-run to place casposon-specific and random substrates of similar type (e.g. TR17 vs ranTR17) in adjacent lanes on the gel. Is a dsran17 substrate (i.e. a fully double-stranded random substrate of 17 bp) also integrated? In addition, ssran12 and ssLE11 could be omitted from this gel, as these substrates are also tested in the rightmost gel in the panel.

7) In paragraph two of the Discussion section, the authors suggest a similarity in the lack of sequence specificity on the bottom strand of the casposase TSD with a lack of sequence specificity for spacer-side integration by Cas1-Cas2. However, it is important to note that Cas1-Cas2 does display specificity for the entire direct repeat sequence, likely through sequence-specific deformations of the repeat that occur during full-site integration (as proposed by Wright et al., 2017). In that study, it was shown that mutations in the repeat affect spacer-side integration, but do not have an effect on leader-side integration. Thus, the conclusion stated by the authors that bottom strand integration is sequence-independent and instead based on a "molecular ruler" is not consistent with current models for spacer-side integration.

Reviewer #1:

Cas1 and Cas2 are hallmarks of the CRISPR-Cas adaptive immune systems found in most archaeal and many bacterial genomes. Koonin and colleagues have previously identified homology between Cas1 and casposon family proteins (doi: 10.1186/1741-7007-12-36) and predicted that CRISPR repeats evolved from the target site duplications (doi: 10.1016/j.mib.2017.04.004). These predictions have been largely validated with in vitro biochemical studies (doi.org/10.1093/nar/gkv1180, doi.org/10.1093/nar/gkw821, doi.org/10.1093/nar/gkz447). Here, Hickman et al., extend the previous work by defining essential features of the target site and testing a wide-variety of integration substrates. This on its own would not warrant publication in *eLife*. However, they use this qualitative biochem to identify substrates that result in higher order assemblies and they determine the structure of the tetrameric complex bound to DNA. Overall the experiments appear technically sound, and the paper delivers the first structure of a casposase. These mechanistic insights and evolutionary connection of this system to CRISPR-Cas immune systems are important and warrant publication in *eLife*. However, the text and figure are a challenge to interpret (esp Figure 1) and considerable revision is required to make this work accessible to a broad readership.

The authors purify a casposases from *Methanosarcina mazei* and determine sequence requirements for integration. The authors describe a sequencing method to determine the location of integration. 7 of 8 integration occurs at the TSD sequence with concomitant generation of a 14 bp TSD. "The other sequenced product arose from integration into a site ~110 bp away from the specific sequence and was also accompanied by a 14 bp TSD." Please include more information. Was there sequence similarity between these sites? If not, what does this say about the mechanism of recognition and how does this relate what the structures teaches us about recognition?

Figure 1A. It is not clear what the authors are intending to teach the reader in panel A. Based on the figure legend title ("Casposase structure and function") the reader is expecting to see a structure, rather than a schematic of the genomic context. Most of the arrows in the schematic are different colors, which have no explanations, the sequence inserted below the schematic is too small to decipher, and the blue box inserted to the right is not labeled or sufficiently explained. To easily understand this figure, readers will need to be familiar with the papers by Koonin et al. I think this is an unfair expectation of readers.

Seems like it would be worth showing a gel or at least referencing the gel in Figure 5—figure supplement 4, when the authors first claim to have "expressed and purified a representative casposase".

Figure 1B. I was unable to understand how the TIR (introduced in font size ~2 in panel B) is different form the TSD, which is introduced in panel A. I understand that it's a tandem inverted repeat but where is the sequence, how long, how is it related to the biology (I'm lost)..? Why is the first gel white on black and the second is black on white? Are these different imaging methods of the same assay? If the 5'-CGCA motif is conserved, then why do the authors mutate the first 6 rather than the first 4 bps of the TSD sequence?

Figure 1C. What are the red dots one end of the orange oligonucleotides?

Figure 2A. In my opinion, seeing a gel of the purified protein is more important to me than seeing a series of SEC runs with increasing concentration of the DNA substrates. If the authors choose to show these in the main text, then a vertical line extending from the protein alone peak down though the other runs will help illustrate the shift. The authors disclose that "the complex did not produce the expected shift on SEC for a tetramer", but the peak does not look symmetrical to me and the left edge appears to be almost vertical. This would suggest that the complex may be eluting in the void, but the x-axis is in time rather than volume, so it is hard to know what this mean for a16/60 Superdex 200 column. Try a Sup 6 column and include a regression line fit with standers. It appears that the labels for the x-axis are misplaced for the last column ("integration products"). In my opinion this entire figure could go in the supplemental material and Figure 1 should be refined and split into two figures.

The text for Table 1 overlaps the table itself making it a little hard to interpret. I suggest adding a sentence to the text that states how the structure was determined (e.g., MR or experimental phases).

Figure 3. Active site subunits "shown in pink sticks" are nearly impossible to see. I could not find an explanation for the green spheres in panel B.

Figure 4. The authors state that "Each casposase monomer exhibits the fold expected of a Cas1 homolog" but none of the figures do a good job of showing this and I could not find an RMSD for a define set of equivalently positioned atoms.

Discussion paragraph five: delete word "is". “It is an intriguing possibility is that some casposases may already possess the ability to integrate short CRISPR-like spacers,…..”

Reviewer #2:

In their manuscript, Hickman and colleagues present the first crystal structure of a transposon-encoded casposase bound to an integration product mimic, revealing a tetrameric architecture in which two Cas1 dimers sandwich the target DNA destined to be duplicated during downstream molecular steps within the pathway. Biochemical experiments are presented to argue that single-stranded transposon ends may be the natural substrate utilized during integration, and that specific sequence motifs in both the transposon end and the target site affect biochemical integration activity. By comparing their casposase-DNA structure to available crystal structures of Cas1 homologs in the context of the Cas1-Cas2 complexes that mediate spacer acquisition during adaptive immunity in CRISPR-Cas systems, the authors argue a plausible evolutionary scenario by which gaining Cas2 interactions could have directed casposases towards an integration mechanism that is more akin to spacer acquisition than transposition.

The crystal structure is a notable advance that broadens our understanding of the Cas1/casposase protein family, and is undoubtedly a substantive contribution to the field. However, in this reviewer's opinion, the biochemical experiments that accompany this finding fall short of providing compelling evidence for a specific integration mechanism, and instead leave more to be desired in terms of what overhang geometries are optimal, how integration sites are chosen, and how casposons are excised. In addition, the authors gloss over the interesting finding that the HTH domain is disordered in the crystals and fail to follow-up with further experiments to investigate the role of this domain on biochemical integration activity, and whether the domain may be important in the context of more physiologically relevant substrates. I believe additional experiments are merited for this study to go beyond reporting a novel structure, to helping yield new mechanistic insights for the process of casposon transposition.

1) Some important controls are missing from the results presented in Figure 1, most notably negative controls in which the plasmid is incubated with casposase in the presence of divalent metal ions but no transposon end substrates. Previous Cas1 studies have observed considerable nicking activity (or possibly nicking/cleavage from contaminating proteins), so it is crucial that the baseline activity of the casposase protein prep on plasmid DNA is evaluated. Furthermore, for the data presented in panels C and D, no lanes exist showing the starting plasmid substrate at time 0 (i.e. in the absence of protein and transposon end substrate); since any given plasmid prep will have variable ratios of supercoiled to open-circular plasmid, it is critical that the reader have this comparison – for the exact substrate used in the experiments – to fully evaluate the biochemical activity of casposase and discriminate this from the starting status of the substrate. I would even argue that naming the bands "target," "SE," and "DE," is misleading, since a plasmid will naturally purify from a bacterial culture with some fraction in the open circular form, which will migrate the same as a SE product but is certainly not a single-end integration product.

2) A major question is left unaddressed (and mostly unmentioned) in the manuscript, namely, in what form the casposon is excised and integrated. I understand this is still unclear in the literature, that previous attempts to monitor excision by this group were unsuccessful (i.e. Hickman and Dyda, 2015), and that there is as yet no in vivo transposition data for casposons. Nevertheless, the reader should certainly be given more context on this topic, in the Introduction/Discussion, and in the Results section to better motivate the range of donor substrates tested. I would also like to see the authors test a larger panel of "Top Strand Recessed" substrates, and to more quantitatively compare integration activities amongst these substrates, and between these substrates and blunt dsDNA substrates. Similar experiments in the context of spacer acquisition by Cas1 complexes (e.g. Wright et al., 2019) have revealed the likely substrates that are used biologically, based on a clear optimum in terms of integration product formation kinetics. The authors have an opportunity to make similar discoveries with their casposase, but the use of just two timepoints, the lack of any product quantification, and the relatively small set of substrates tested, limits the potential of their biochemical assays.

3) Related to #2, I would prefer more experimental support and/or justification for the specific choice of the 8-nt overhang integration product. Why was 8 selected as the overhang length? Was a larger crystallization screen performed, and this was the only substrate geometry that resulted in diffracting crystals? Is there any further evidence that this is a physiologically relevant ssDNA end sequence? What happens if the casposon end substrates are shorter, or partially double-stranded? For example, it seems that the iSCAMS analysis could be scaled up to test a wider range of model integration products, to investigate whether tetramer formation is more or less favored with certain substrates, which could help suggest what may or may not be more 'relevant.'

4) The authors point out that capsposases have an additional C-terminal HTH domain that is absent in Cas1, and that this domain could not be modeled into the experimental electron density. In light of these interesting observations, the authors should test the biochemical activity casposase mutants in which this domain is truncated; does these mutants show behave similarly? Differently? Although it would be interesting to further investigate whether this domain becomes ordered with different DNA substrates, I understand that pursuing additional structures may be too much to ask. However, certainly some biochemical investigations are warranted here. I would also caution the authors against stating that the HTH domain "is not in a folded state" – although I am not a structural biologist, the absence of ordered density in a X-ray crystal structure is not equivalent to a domain being unfolded.

5) A major take-away from this manuscript would be that this casposase integrates substrates primarily at a specific target site rather than randomly, as has been previously touched upon in the literature. However, apparent integration events in pUC19 alone and farther away from the target site have been detected in this and previous studies. No sequencing data for these events are shown, and the authors failed to perform additional analysis on integration patterns. It has not become routine in spacer acquisition studies with Cas1 to deep sequence half-site integration products in order to gain deeper insights into specificity, and the authors should perform similar NGS experiments here to enhance their biochemical results. These experiments could provide much more data on the target site requirements, but looking for conservation (or lack of conservation) in the motifs that are preferred adjacent to integration sites other than at the 3' end of tRNA-Leu.

Reviewer #3:

In this study, Hickman et al. describe the activity and structure of a casposase, the hypothesized evolutionary ancestor of the Cas1 integrase found in CRISPR-Cas adaptive immune systems. The authors demonstrate in vitro DNA integration by the casposase from *Methanosarcina mazei*, and perform biochemical experiments to establish the target sequence specificity and substrate preferences for this casposase. They also report, for the first time, a structure of the casposase bound to an integration product mimic. The casposase forms a tetramer composed of two casposase dimers sandwiching the DNA. The structure provides insight into the mechanism of substrate and target recognition by the casposase. In addition, comparison of the casposase and Cas1-Cas2 complex structures show that the casposase tetramer is mutually exclusive with Cas2 binding, and suggests how architectural changes in the Cas1-Cas2 complex may have led to mechanistic differences between the two types of integrases. The structure of the casposase is an important first step in understanding casposon function and the evolution of CRISPR-Cas adaptation mechanisms, and the authors propose several interesting hypotheses based on the structure. However, these hypotheses are largely speculative and could greatly benefit from experimental evidence. In particular, the structural data is not supported by any biochemical experiments to establish a structure-function relationship. Thus, the structure is largely descriptive, although the authors speculate on the importance of a number of observed interactions within the structure. Several examples are provided in comment 1 below, in addition to few other concerns that should also be addressed by the authors.

1) When describing the structure, the authors note several protein DNA contacts, but it is unclear whether any of these contacts contribute to the specificity for the TSD target or LE substrate. From the data shown in Figure 1C, it appears that the casposase is somewhat specific for the LE sequence, especially for shorter single-stranded DNAs. Can this specificity be explained by the structure? Do R191 and R192 contribute to the specificity of the casposase for the TSD? The authors speculate about the importance of stacking interactions for ssDNA substrate binding (subsection “ssDNA casposon end binding resembles that of spacer 3' overhangs”). Did the authors perform any mutagenesis experiments to establish the importance of these contacts for target and substrate specificity? In general, the study would be greatly improved by experiments supporting the structure-based hypotheses proposed by the authors when describing the structure.

2) The order in which substrates are shown on the left gel in Figure 1C is confusing. If possible, it would be helpful if this gel could be re-run to place casposon-specific and random substrates of similar type (e.g. TR17 vs ranTR17) in adjacent lanes on the gel. Is a dsran17 substrate (i.e. a fully double-stranded random substrate of 17 bp) also integrated? In addition, ssran12 and ssLE11 could be omitted from this gel, as these substrates are also tested in the rightmost gel in the panel.

3) In paragraph two of the Discussion section, the authors suggest a similarity in the lack of sequence specificity on the bottom strand of the casposase TSD with a lack of sequence specificity for spacer-side integration by Cas1-Cas2. However, it is important to note that Cas1-Cas2 does display specificity for the entire direct repeat sequence, likely through sequence-specific deformations of the repeat that occur during full-site integration (as proposed by Wright et al., 2017). In that study, it was shown that mutations in the repeat affect spacer-side integration, but do not have an effect on leader-side integration. Thus, the conclusion stated by the authors that bottom strand integration is sequence-independent and instead based on a "molecular ruler" is not consistent with current models for spacer-side integration.

[Editors’ note: further revisions were suggested prior to acceptance, as described below.]

Thank you for resubmitting your work entitled "Casposase structure and the mechanistic link between DNA transposition and spacer acquisition by CRISPR-Cas" for further consideration by *eLife*. Your revised article has been evaluated by John Kuriyan (Senior Editor) and a Reviewing Editor.

The reviewers agree that the manuscript has been dramatically improved and there is unanimous consensus that the paper should be published in *eLife*. The reviewers offer a few additional suggestions, as outlined below:

Summary:

In this manuscript, Hickman et al. express and purify a casposase from *Methanosarsina mazei*. They demonstrate that the MmCasposase integrates a wide variety of RNA and DNA substrates into a specific location. Integration requires substrates with a free 3'-OH and a target that contains specific sequence motifs. To understand the mechanism of integration the authors attempt to crystallize the casposase bound to DNA oligonucleotides representing different stages of the integration reaction. Diffracting crystals of the casposase bound to a substrate that mimics the post integration step were used to determine a 3.1 Å resolution structure. The structure reveals a tetrameric architecture in which two casposase dimers sandwich the target DNA. The authors highlight several interactions with target and the substrate, and mutants are used to test the importance of these interactions. The authors compare this structure to available crystal structures of Cas1 and Cas1-2 complexes. This analysis shows that the casposase tetramer is mutually exclusive with Cas2 binding and suggests how architectural changes in the Cas1-Cas2 complex may have led to mechanistic differences between the two types of integrases. Collectively, the biochemistry and structure presented support a speculative discussion that connects architectural changes in the casposase that would accommodate Cas2 binding and the transition to a Cas1-Cas2 complex that is necessary for spacer integration. While clarity of presentation could be improved in places, the work is rigorously performed, and the conclusions are of interest to a broad readership.

Organization of the presentation:

The initial characterization of casposon substrate and target specificities is much improved by the new experiments. However, the presentation is a bit disjointed, as it jumps back and forth between characterization of substrate and target requirements. For example, Figures 1C, 3A-B and 3D all investigate casposon end requirements, while Figures 2 and 3C are on target specificity. Indeed, in one case this requires that the reader read ahead to understand the experiment explained in Figure 2 ("for the rationale, see Figure 3A and associated discussion"). This section could flow better if the authors combine Figure 1C, 3A-B, and 3D into a new Figure, which could be Figure 2. This would also allow the authors to show a panel describing all substrates used in these three panels, as it is currently confusing that TR17 and ssLE17 are shown in Figure 1C but not mentioned until much later in the text. Figures 2 and 3C could then be combined to form a new Figure 3. With these changes, the authors could first define casposon end requirements, and then describe the characterization of the target sequence.

---

## [Author Response]

[Editors’ note: what follows is the authors’ response to the first round of review]

In this manuscript, Hickman et al. present the first crystal structure of a transposon-encoded casposase bound to an integration product mimic. The structure reveals a tetrameric architecture in which two casposase dimers sandwich the target DNA. The authors compare this structure to available crystal structures of Cas1 and Cas1-2 complexes. This analysis shows that the casposase tetramer is mutually exclusive with Cas2 binding and suggests how architectural changes in the Cas1-Cas2 complex may have led to mechanistic differences between the two types of integrases.The reviewers all agree that the structure presented in this manuscript is potentially important. However, the structural data are not supported by biochemical experiments that advance our understanding of mechanism.Some of the specific points raised during the review and in the discussion between reviewers and the editors are noted below, and the individual reviews are appended. We concluded that it could take substantial time to address all of these issues in a satisfactory manner. At this point, the authors may simply wish to submit this high quality paper elsewhere, without delay. If, however, these points can be addressed satisfactorily then the paper could be resubmitted to eLife for further consideration, should the authors wish to do so.1) Clarification on the DNA substrates used in the biochemical assay, and an explanation for how these substrates relate to the biology are required.

We have repeated all of the biochemical assays and incorporated more controls and time points (see below). In doing so, we took the opportunity to perform the biochemical assay first discussed in the text using an authentic *M. mazei* casposon 31mer TIR (Figure 1B). In the text, we now reference the work on CRISPR Cas1-Cas2 proteins to explain our exploration of substrates with 3'-OH overhangs. The use of single-stranded substrates was the outcome of performing control reactions (or what we thought would be control reactions), and their activity was unexpected. Since then, Krupovic et al. (2019) have reported that a single-stranded casposon end is an effective substrate for the *A. boonei* casposase, and we have cited their work. Unfortunately, as there are many unknowns regarding the experimental biology for casposons, we can only speculate as to how these particular substrates relate to their biology.

Additional controls and additional time points are necessary. Important controls are missing from the results presented in Figure 1, most notably negative controls in which the plasmid is incubated with casposase in the presence of divalent metal ions but no transposon end substrates. Previous Cas1 studies have observed considerable nicking activity (or possibly nicking/cleavage from contaminating proteins), so it is crucial that the baseline activity of the casposase protein prep on plasmid DNA is evaluated. Furthermore, for the data presented in panels C and D, no lanes exist showing the starting plasmid substrate at time 0 (i.e. in the absence of protein and transposon end substrate); since any given plasmid prep will have variable ratios of supercoiled to open-circular plasmid, it is critical that the reader have this comparison – for the exact substrate used in the experiments – to fully evaluate the biochemical activity of casposase and discriminate this from the starting status of the substrate. Labeling the bands "target," "SE," and "DE," is misleading, since a plasmid will naturally purify from a bacterial culture with some fraction in the open circular form, which will migrate the same as a SE product but is certainly not a single-end integration product.

We have repeated the experiments presented in the original manuscript (including those described in Figure 1) and included additional controls and time points. In our revised manuscript, for all experiments involving plasmid substrates, a negative control is shown in which plasmid is incubated with the casposase and metal ion in the absence of a transposon end substrate. We have re-labeled the product band positions as suggested by the reviewers, and these are now indicated as "relaxed", and "linear". The target plasmid is now labelled "sc". We have retained the SE and DE terminology in the schematic in Figure 1B, as these are the products expected by the model. As a reader can now judge, there is not much nonspecific nicking with our purified casposase under the assay conditions used.

2) Provide more experimental support and/or justification for the specific choice of the 8-nt overhang integration product. Why was 8 selected as the overhang length? Was a larger crystallization screen performed, and this was the only substrate geometry that resulted in diffracting crystals? Is there any further evidence that this is a physiologically relevant ssDNA end sequence?

We apologize, as in our original submission we failed to mention that we tried a wide range of DNA oligonucleotides for structural studies over the course of many months, and that – to date – the only substrate that has yielded crystals that diffract X-rays well enough to allow us to solve the structure is the 8-nt overhang integration product. We believe that such exploration (riddled with many failures) is typical in experimental structural biology. As shown in Figure 3B, an 8-nt ssDNA casposon sequence produces products in an integration assay that are specific for the target site (i.e., they are not seen with pUC19 alone). Thus, it has the specificity expected for a biologically relevant end sequence but we cannot say more than this in the absence of an in vivo assay for casposon mobility.

For at least one reviewer, it was unclear what the SEC data provided that was not show by the iSCAMS analysis. All reviewers commented on misplaced labels and inappropriate units (time rather than volume) in the SEC. There was some discussion about moving the SEC to the supplement but regardless of where it is discussed, the interpretation of these data must be clarified.

We agree with the reviewer that the SEC data are perhaps not as important in light of the iSCAMS analysis; therefore we have chosen to present it (and some new data) now as supplemental material (Figure 4—figure supplement 1 and 2). The supplemental figures now show the chromatograms with corrected labels and volume units. We have been able to clarify the apparent disconnect between the iSCAMS and SEC results with further SEC experiments in which we performed a dilution series with the protein alone and complexes with TR17 and the crystallized integration product. These revealed that the elution properties of the casposase alone and its complexes change as a function of protein concentration. We conclude that the lower concentration used for iSCAMS more accurately reflects the property of the casposase in vivo. These results are now discussed in the revised text.

3) The authors gloss over the interesting finding that the HTH domain is disordered in the crystals and fail to follow-up with further experiments to investigate the role of this domain on biochemical integration activity, and whether the domain may be important in the context of more physiologically relevant substrates.

We used the structure as a guide to produce a soluble version of the casposase missing the C-terminal domain (residues 1-341, "ΔHTH") – solubility being a property that had frustratingly eluded us in the absence of structural information. We have therefore been able to follow up with integration experiments using a range of substrates, and these results are now presented in Figure 6 and associated text. Indeed, the C-terminal domain is crucial, although we do not know if it is recognizing the oligonucleotide end or the target. To date, we have not been able to confirm an interaction with either.

4) A major take-away from this manuscript would be that this casposase integrates substrates primarily at a specific target site rather than randomly, as has been previously touched upon in the literature. However, apparent integration events in pUC19 alone and farther away from the target site have been detected in this and previous studies. No sequencing data for these events are shown, and the authors fail to perform additional analysis on integration patterns. The sequences should be shown and discuss. It has become routine in spacer acquisition studies with Cas1 to deep sequence half-site integration products in order to gain deeper insights into specificity. Similar NGS experiments are warranted here to enhance their biochemical results. While NGS experiments are not required for publication in eLife, the authors should perform bioinforamtics analyses to determine whether related casposons are always present in genomes adjacent to similar target sites. Please mention, in the Discussion, the recent observation that the Type V-C Cas1 is much less sequence-specific in its integration site preference.

We agree with the reviewer's suggestion that the need for an NGS experiment was justified. We recruited the expertise of our colleagues, Dr. Pavel and Dr. Haase, who helped us to perform the NGS analysis on the linear product of targeted integration. They are now included as co-authors, and the new data are presented in Figure 2 and discussed in the text. As suggested, we have added a sentence regarding Type V-C Cas1 to the Discussion as follows: “Finally, the V-C Cas1 is less sequence-specific in its integration site preference relative to other characterized Cas1-Cas2 integrases (Wright et al., 2019), in marked contrast to the stringent sequence-specificity we observed here for the *M. mazei* casposase (Figure 2).”

5) When describing the structure, the authors note several protein-DNA contacts, but it is unclear whether any of these contacts contribute to the specificity for the TSD target or LE substrate. From the data shown in Figure 1C, it appears that the casposase is somewhat specific for the LE sequence, especially for shorter single-stranded DNAs. Can this specificity be explained by the structure? Do R191 and R192 contribute to the specificity of the casposase for the TSD? The authors speculate about the importance of stacking interactions for ssDNA substrate binding (subsection “ssDNA casposon end binding resembles that of spacer 3' overhangs”). Did the authors perform any mutagenesis experiments to establish the importance of these contacts for target and substrate specificity? Experiments supporting structure-based hypotheses proposed by the authors are required.

As suggested, in order to explain the specificity for short ssDNAs, we generated several mutants to test the roles of the residues observed to form interactions with the ssDNA. These include three mutant versions of the casposase that the structure predicted should disrupt the stacking interactions, as well as the double-mutant R191A/R192A. We have compared their activity on the TSD target vs. pUC19 to evaluate target specificity, and have tested their integration activity using a range of substrates to probe the role of these residues in substrate specificity (Figure 7F and Figure 7—figure supplement 1). These data are now discussed in the text.

6) The order in which substrates are shown on the left gel in Figure 1C is confusing. If possible, it would be helpful if this gel could be re-run to place casposon-specific and random substrates of similar type (e.g. TR17 vs ranTR17) in adjacent lanes on the gel. Is a dsran17 substrate (i.e. a fully double-stranded random substrate of 17 bp) also integrated? In addition, ssran12 and ssLE11 could be omitted from this gel, as these substrates are also tested in the rightmost gel in the panel.

We apologize for the haphazard organization of the gels presented in our original submission. As suggested, we have repeated the experiment to include a dsran17 substrate and the order of the lanes has been changed. These data are now in Figure 3A.

7) In paragraph two of the Discussion section, the authors suggest a similarity in the lack of sequence specificity on the bottom strand of the casposase TSD with a lack of sequence specificity for spacer-side integration by Cas1-Cas2. However, it is important to note that Cas1-Cas2 does display specificity for the entire direct repeat sequence, likely through sequence-specific deformations of the repeat that occur during full-site integration (as proposed by Wright et al., 2017). In that study, it was shown that mutations in the repeat affect spacer-side integration, but do not have an effect on leader-side integration. Thus, the conclusion stated by the authors that bottom strand integration is sequence-independent and instead based on a "molecular ruler" is not consistent with current models for spacer-side integration.

Upon repeating our initial experiments with time courses, it has become clear that our original statement has to be revised. As shown in Figure 3C, when the activity of the "mutR" substrate is compared to that of "symTSD", two substrates that differ in sequence at the site of "bottom strand" integration, there is clearly a difference. We have removed the entire section from the revised text.

Reviewer #1:Cas1 and Cas2 are hallmarks of the CRISPR-Cas adaptive immune systems found in most archaeal and many bacterial genomes. Koonin and colleagues have previously identified homology between Cas1 and casposon family proteins (doi: 10.1186/1741-7007-12-36) and predicted that CRISPR repeats evolved from the target site duplications (doi: 10.1016/j.mib.2017.04.004). These predictions have been largely validated with in vitro biochemical studies (doi.org/10.1093/nar/gkv1180, doi.org/10.1093/nar/gkw821, doi.org/10.1093/nar/gkz447). Here, Hickman et al., extend the previous work by defining essential features of the target site and testing a wide-variety of integration substrates. This on its own would not warrant publication in eLife. However, they use this qualitative biochem to identify substrates that result in higher order assemblies and they determine the structure of the tetrameric complex bound to DNA. Overall the experiments appear technically sound, and the paper delivers the first structure of a casposase. These mechanistic insights and evolutionary connection of this system to CRISPR-Cas immune systems are important and warrant publication in eLife. However, the text and figure are a challenge to interpret (esp Figure 1) and considerable revision is required to make this work accessible to a broad readership.The authors purify a casposases from Methanosarcina mazei and determine sequence requirements for integration. The authors describe a sequencing method to determine the location of integration. 7 of 8 integration occurs at the TSD sequence with concomitant generation of a 14 bp TSD. "The other sequenced product arose from integration into a site ~110 bp away from the specific sequence and was also accompanied by a 14 bp TSD." Please include more information. Was there sequence similarity between these sites? If not, what does this say about the mechanism of recognition and how does this relate what the structures teaches us about recognition?

We have now performed next-generation sequencing on the linear product of the plasmid integration reaction, and included these results in the revised manuscript. As shown in Figure 2C, there is indeed sequence similarity between the favored integration site and the very few other detected sites of integration.

Figure 1A. It is not clear what the authors are intending to teach the reader in panel A. Based on the figure legend title ("Casposase structure and function") the reader is expecting to see a structure, rather than a schematic of the genomic context. Most of the arrows in the schematic are different colors, which have no explanations, the sequence inserted below the schematic is too small to decipher, and the blue box inserted to the right is not labeled or sufficiently explained. To easily understand this figure, readers will need to be familiar with the papers by Koonin et al. I think this is an unfair expectation of readers.

In light of these comments, we have substantially revised Figure 1A, and have included more labeling, increased the font size for the TSD sequence, and have included the sequence of the 31mer TIR. We have also added titles to the two figure sections. The figure legend title has also been changed to: "*Methanosarcina mazei* casposon organization and initial biochemical characterization."

Seems like it would be worth showing a gel or at least referencing the gel in Figure 5—figure supplement 4, when the authors first claim to have "expressed and purified a representative casposase".

In the revised manuscript, in addition to the gel in the supplemental figure showing dissolved crystals and the protein control (Figure 5—figure supplement 4), we now show a gel for all of the purified proteins used in this study in Figure 6A.

Figure 1B. I was unable to understand how the TIR (introduced in font size ~2 in panel B) is different form the TSD, which is introduced in panel A. I understand that it's a tandem inverted repeat but where is the sequence, how long, how is it related to the biology (I'm lost)..?

We have now included a better schematic including the entire relevant genomic sequence in Figure 1A.

Why is the first gel white on black and the second is black on white?

We thank the reviewer for pointing this out and, in fact, there was no good reason. This has been corrected in the revised figures.

Are these different imaging methods of the same assay?

The gels have now been standardized to black-on-white throughout the manuscript.

If the 5'-CGCA motif is conserved, then why do the authors mutate the first 6 rather than the first 4 bps of the TSD sequence?

The experiment shown in the original submission was performed before we recognized that the first 4 bp are important, not 6. We have therefore removed this experiment panel from Figure 1B; instead, we have tested the effect of mutating only the first 4 bp of the TSD sequence in the experiment shown in Figure 3C.

Figure 1C. What are the red dots one end of the orange oligonucleotides?

The red dot indicates the 3'-OH that is the nucleophile for the integration reaction, and this is now stated in the figure legend for the schematic in Figure 1B. We have replaced the original schematic involving red dots with the explicit sequences of the oligonucleotides used (Figure 3C).

Figure 2A. In my opinion, seeing a gel of the purified protein is more important to me than seeing a series of SEC runs with increasing concentration of the DNA substrates.

We agree with the reviewer, and an SDS-PAGE gel is now shown in Figure 6A (in addition to the control lane in Figure 5—figure supplement 4), and the SEC data are now in supplementary material (Figure 4—figure supplement 1 and 2).

If the authors choose to show these in the main text, then a vertical line extending from the protein alone peak down though the other runs will help illustrate the shift.

We appreciate the suggestion, and vertical lines are now incorporated.

The authors disclose that "the complex did not produce the expected shift on SEC for a tetramer", but the peak does not look symmetrical to me and the left edge appears to be almost vertical. This would suggest that the complex may be eluting in the void, but the x-axis is in time rather than volume, so it is hard to know what this mean for a16/60 Superdex 200 column.

To resolve these issues, we performed further SEC experiments over a range of protein concentrations and generated a standard curve for the Superdex 200 Increase 3.2/300 column to allow us to estimate MWs. Although the peak shape is indeed asymmetric, at three different concentrations the integration product complex eluted with a peak position consistent with a tetramer bound to one DNA substrate.

Try a Sup 6 column and include a regression line fit with standers.

We believe we have resolved the basis for our original confusion (we had not recognized the effect of protein concentration on elution position), so have not repeated the experiment on a Superose 6 column. The regression line fit is now shown in Figure 4—figure supplement 2.

It appears that the labels for the x-axis are misplaced for the last column ("integration products").

Thank you for pointing this out, and the figure has been corrected.

In my opinion this entire figure could go in the supplemental material and Figure 1 should be refined and split into two figures.

We agree, and this has been done.

The text for Table 1 overlaps the table itself making it a little hard to interpret. I suggest adding a sentence to the text that states how the structure was determined (e.g., MR or experimental phases).

As suggested, we have included the following when we first introduce the structure: “…was solved to 3.1 Å using experimental phases obtained from a selenomethionine derivative.” The text for the table has been corrected.

Figure 3. Active site subunits "shown in pink sticks" are nearly impossible to see. I could not find an explanation for the green spheres in panel B.

The active site is now shown in a close-up view (Figure 5C), and an explanation for the green spheres (Ca^2+^ ions from the crystallization buffer) has been included in the figure legend to Figure 5B.

Figure 4. The authors state that "Each casposase monomer exhibits the fold expected of a Cas1 homolog" but none of the figures do a good job of showing this and I could not find an RMSD for a define set of equivalently positioned atoms.

We have added a figure (Figure 5—figure supplement 2) that shows the side-by-side structures of the casposase dimer and that of Cas1 from *A. fulgidus*, as well as those of the aligned N-termini and catalytic domains. The following sentence has now been included in the text: "When compared to its most closely related structurally characterized Cas1 homolog, that from *Archaeoglobus fulgidus* (Kim et al., 2013) with which it shares 32% sequence identity, the N-terminal domains have an RMSD of 1.68 Å over 72 shared Cα positions and the catalytic domains are even more similar, with an RMSD of 1.28 Å over 208 shared Cα positions."

Discussion paragraph five: delete word "is". “It is an intriguing possibility is that some casposases may already possess the ability to integrate short CRISPR-like spacers,…..”

Done – thank you.

Reviewer #2:In their manuscript, Hickman and colleagues present the first crystal structure of a transposon-encoded casposase bound to an integration product mimic, revealing a tetrameric architecture in which two Cas1 dimers sandwich the target DNA destined to be duplicated during downstream molecular steps within the pathway. Biochemical experiments are presented to argue that single-stranded transposon ends may be the natural substrate utilized during integration, and that specific sequence motifs in both the transposon end and the target site affect biochemical integration activity. By comparing their casposase-DNA structure to available crystal structures of Cas1 homologs in the context of the Cas1-Cas2 complexes that mediate spacer acquisition during adaptive immunity in CRISPR-Cas systems, the authors argue a plausible evolutionary scenario by which gaining Cas2 interactions could have directed casposases towards an integration mechanism that is more akin to spacer acquisition than transposition.The crystal structure is a notable advance that broadens our understanding of the Cas1/casposase protein family, and is undoubtedly a substantive contribution to the field. However, in this reviewer's opinion, the biochemical experiments that accompany this finding fall short of providing compelling evidence for a specific integration mechanism, and instead leave more to be desired in terms of what overhang geometries are optimal, how integration sites are chosen, and how casposons are excised. In addition, the authors gloss over the interesting finding that the HTH domain is disordered in the crystals and fail to follow-up with further experiments to investigate the role of this domain on biochemical integration activity, and whether the domain may be important in the context of more physiologically relevant substrates. I believe additional experiments are merited for this study to go beyond reporting a novel structure, to helping yield new mechanistic insights for the process of casposon transposition.1) Some important controls are missing from the results presented in Figure 1, most notably negative controls in which the plasmid is incubated with casposase in the presence of divalent metal ions but no transposon end substrates. Previous Cas1 studies have observed considerable nicking activity (or possibly nicking/cleavage from contaminating proteins), so it is crucial that the baseline activity of the casposase protein prep on plasmid DNA is evaluated. Furthermore, for the data presented in panels C and D, no lanes exist showing the starting plasmid substrate at time 0 (i.e. in the absence of protein and transposon end substrate); since any given plasmid prep will have variable ratios of supercoiled to open-circular plasmid, it is critical that the reader have this comparison – for the exact substrate used in the experiments – to fully evaluate the biochemical activity of casposase and discriminate this from the starting status of the substrate. I would even argue that naming the bands "target," "SE," and "DE," is misleading, since a plasmid will naturally purify from a bacterial culture with some fraction in the open circular form, which will migrate the same as a SE product but is certainly not a single-end integration product.

These issues have been addressed by repeating the experiments in Figure 1, as discussed above.

2) A major question is left unaddressed (and mostly unmentioned) in the manuscript, namely, in what form the casposon is excised and integrated. I understand this is still unclear in the literature, that previous attempts to monitor excision by this group were unsuccessful (i.e. Hickman and Dyda, 2015), and that there is as yet no in vivo transposition data for casposons. Nevertheless, the reader should certainly be given more context on this topic, in the Introduction/Discussion, and in the Results section to better motivate the range of donor substrates tested.

We agree with the reviewer that the excision of casposons is an open question at this point. However, one possible consequence of the model for self-synthesizing transposons is that excision may not in fact be occurring as the element might be synthesized at the donor site without the need for excision. However, this is pure speculation at this point. We have included a few sentences in the revised Introduction in which we discuss how free casposon ends might be generated.

I would also like to see the authors test a larger panel of "Top Strand Recessed" substrates, and to more quantitatively compare integration activities amongst these substrates, and between these substrates and blunt dsDNA substrates. Similar experiments in the context of spacer acquisition by Cas1 complexes (e.g. Wright et al., 2019) have revealed the likely substrates that are used biologically, based on a clear optimum in terms of integration product formation kinetics. The authors have an opportunity to make similar discoveries with their casposase, but the use of just two timepoints, the lack of any product quantification, and the relatively small set of substrates tested, limits the potential of their biochemical assays.

As suggested, we have performed the plasmid integration assay with a larger panel of "Top Strand Recessed" substrates, ranging from blunt-ended to 7-nt recessed (Figure 3D). We have also performed the assay to compare blunt dsDNA, recessed, and ssDNA (Figure 3A). We also repeated the assay in triplicate in order to provide a quantitative comparison of their integration activities. These results are plotted in Figure 3—figure supplement 1.

3) Related to #2, I would prefer more experimental support and/or justification for the specific choice of the 8-nt overhang integration product. Why was 8 selected as the overhang length? Was a larger crystallization screen performed, and this was the only substrate geometry that resulted in diffracting crystals? Is there any further evidence that this is a physiologically relevant ssDNA end sequence? What happens if the casposon end substrates are shorter, or partially double-stranded?

As discussed above, we have tried a wide range of substrates in crystallization experiments and with the exception of the 8-nt overhang, none has yielded crystals of sufficient diffraction quality. We have added the following so the reader can understand why this particular complex was studied: "In an effort to further characterize the casposase and its interactions with its substrates, we attempted to crystallize the casposase bound to DNA oligonucleotides representing different stages of the integration reaction. Many casposase-DNA complexes yielded poorly-diffracting crystals including the casposase bound to recessed dsDNA oligonucleotides and ssDNA oligonucleotides of varying length representing the casposon ends. This was also the case with casposase bound to mimics of the post-integration step using both symmetric and asymmetric complete integration target sites and casposon ends of varying length and dsDNA-ssDNA composition. We were eventually able to obtain diffracting crystals of the casposase…"

For example, it seems that the iSCAMS analysis could be scaled up to test a wider range of model integration products, to investigate whether tetramer formation is more or less favored with certain substrates, which could help suggest what may or may not be more 'relevant.'

We agree with the reviewer, and this line of inquiry is one of our future plans.

4) The authors point out that capsposases have an additional C-terminal HTH domain that is absent in Cas1, and that this domain could not be modeled into the experimental electron density. In light of these interesting observations, the authors should test the biochemical activity casposase mutants in which this domain is truncated; does these mutants show behave similarly? Differently?

As discussed above, we have performed these experiments with a 1-341 construct (ΔHTH) and they are shown in Figure 6B. We note that we have not been able to express a range of truncated casposase mutants, as all other deletion constructs we generated prior to the structure determination were insoluble. In our hands, only Δ341 and Δ343 have been soluble.

Although it would be interesting to further investigate whether this domain becomes ordered with different DNA substrates, I understand that pursuing additional structures may be too much to ask.

Our pursuit of additional structures by X-ray crystallography is ongoing. However, in light of our experience to date with the difficulties of obtaining diffraction-quality crystals with interesting DNA constructs, we are also pursuing single particle cryo-EM experiments.

However, certainly some biochemical investigations are warranted here. I would also caution the authors against stating that the HTH domain "is not in a folded state" – although I am not a structural biologist, the absence of ordered density in a X-ray crystal structure is not equivalent to a domain being unfolded.

The reviewer is correct, and we have reworded this to indicate that "the C-terminal residues do not appear to be folded into a compact domain."

5) A major take-away from this manuscript would be that this casposase integrates substrates primarily at a specific target site rather than randomly, as has been previously touched upon in the literature. However, apparent integration events in pUC19 alone and farther away from the target site have been detected in this and previous studies. No sequencing data for these events are shown, and the authors failed to perform additional analysis on integration patterns. It has not become routine in spacer acquisition studies with Cas1 to deep sequence half-site integration products in order to gain deeper insights into specificity, and the authors should perform similar NGS experiments here to enhance their biochemical results. These experiments could provide much more data on the target site requirements, but looking for conservation (or lack of conservation) in the motifs that are preferred adjacent to integration sites other than at the 3' end of tRNA-Leu.

As mentioned above, NGS experiments have been performed and the results discussed in the text and shown in Figure 2.

Reviewer #3:In this study, Hickman et al. describe the activity and structure of a casposase, the hypothesized evolutionary ancestor of the Cas1 integrase found in CRISPR-Cas adaptive immune systems. The authors demonstrate in vitro DNA integration by the casposase from Methanosarcina mazei, and perform biochemical experiments to establish the target sequence specificity and substrate preferences for this casposase. They also report, for the first time, a structure of the casposase bound to an integration product mimic. The casposase forms a tetramer composed of two casposase dimers sandwiching the DNA. The structure provides insight into the mechanism of substrate and target recognition by the casposase. In addition, comparison of the casposase and Cas1-Cas2 complex structures show that the casposase tetramer is mutually exclusive with Cas2 binding, and suggests how architectural changes in the Cas1-Cas2 complex may have led to mechanistic differences between the two types of integrases. The structure of the casposase is an important first step in understanding casposon function and the evolution of CRISPR-Cas adaptation mechanisms, and the authors propose several interesting hypotheses based on the structure. However, these hypotheses are largely speculative and could greatly benefit from experimental evidence. In particular, the structural data is not supported by any biochemical experiments to establish a structure-function relationship. Thus, the structure is largely descriptive, although the authors speculate on the importance of a number of observed interactions within the structure. Several examples are provided in comment 1 below, in addition to few other concerns that should also be addressed by the authors.1) When describing the structure, the authors note several protein DNA contacts, but it is unclear whether any of these contacts contribute to the specificity for the TSD target or LE substrate. From the data shown in Figure 1C, it appears that the casposase is somewhat specific for the LE sequence, especially for shorter single-stranded DNAs. Can this specificity be explained by the structure? Do R191 and R192 contribute to the specificity of the casposase for the TSD? The authors speculate about the importance of stacking interactions for ssDNA substrate binding (subsection “ssDNA casposon end binding resembles that of spacer 3' overhangs”). Did the authors perform any mutagenesis experiments to establish the importance of these contacts for target and substrate specificity? In general, the study would be greatly improved by experiments supporting the structure-based hypotheses proposed by the authors when describing the structure.

As discussed above, we have now performed these experiments and the results are shown in Figure 7F and Figure 7—figure supplement 1.

2) The order in which substrates are shown on the left gel in Figure 1C is confusing. If possible, it would be helpful if this gel could be re-run to place casposon-specific and random substrates of similar type (e.g. TR17 vs ranTR17) in adjacent lanes on the gel. Is a dsran17 substrate (i.e. a fully double-stranded random substrate of 17 bp) also integrated? In addition, ssran12 and ssLE11 could be omitted from this gel, as these substrates are also tested in the rightmost gel in the panel.

We have rerun the assay to both expand the range of substrates tested and to place them in a logical order. The results are shown in Figure 3A.

3) In paragraph two of the Discussion section, the authors suggest a similarity in the lack of sequence specificity on the bottom strand of the casposase TSD with a lack of sequence specificity for spacer-side integration by Cas1-Cas2. However, it is important to note that Cas1-Cas2 does display specificity for the entire direct repeat sequence, likely through sequence-specific deformations of the repeat that occur during full-site integration (as proposed by Wright et al., 2017). In that study, it was shown that mutations in the repeat affect spacer-side integration, but do not have an effect on leader-side integration. Thus, the conclusion stated by the authors that bottom strand integration is sequence-independent and instead based on a "molecular ruler" is not consistent with current models for spacer-side integration.

We thank the reviewer for pointing this out. As discussed above, upon further examination of our data, we realize that our original statement was not correct. We have removed the entire section from the revised manuscript.

[Editors’ note: what follows is the authors’ response to the second round of review.]

Organization of the presentation:The initial characterization of casposon substrate and target specificities is much improved by the new experiments. However, the presentation is a bit disjointed, as it jumps back and forth between characterization of substrate and target requirements. For example, Figures 1C, 3A-B and 3D all investigate casposon end requirements, while Figures 2 and 3C are on target specificity. Indeed, in one case this requires that the reader read ahead to understand the experiment explained in Figure 2 ("for the rationale, see Figure 3A and associated discussion"). This section could flow better if the authors combine Figure 1C, 3A-B, and 3D into a new Figure, which could be Figure 2. This would also allow the authors to show a panel describing all substrates used in these three panels, as it is currently confusing that TR17 and ssLE17 are shown in Figure 1C but not mentioned until much later in the text. Figures 2 and 3C could then be combined to form a new Figure 3. With these changes, the authors could first define casposon end requirements, and then describe the characterization of the target sequence.

We agree that most of these suggestions would help with the logic and the flow. However, a minor problem arises with Figure 3D which is certainly related to substrate specificity (as Figure 1C, 3A-B) but uses an assay that is introduced and validated only in Figure 3C (i.e., substrates are integrated into an oligo targ40 but not ran40). The solution we chose is to keep 3C and 3D together. We have therefore created a new Figure 2 with 1C, 3A-B, and combined old Figures 2 and 3C-3D into new Figure 3.